# Revisiting the African mtDNA landscape through complete mitochondrial genomes

Imke Lankheet [1], Afifa Chowdhury [1], Christian Tellgren-Roth [2,3], Cécile Jolly[1], André E. R. Soares [4], Miguel de Navascués [5], Sara Pacchiarotti [6], Lorenzo Maselli [6,7,8], Guy Kouarata[6], Jean-Pierre Donzo[6,9], Vinet Coetzee[10], Minique de Castro[11], Peter Ebbesen[12], Edita Priehodová[13], Eliška Podgorná[13], Viktor Černý[13], Susanne T. Green[14], Pakou Harena[15], Lebarama Bakrobena[15], Forka Leypey Mathew Fomine[16], Zelalem GebreMariam Tolesa [17], Wendawek Abebe Mengesha [17], Michael de Jongh[18], Himla Soodyall[19,20], Koen Bostoen [6], Chiara Barbieri [21,22], Maximilian Larena [1], Helena Malmström [1] & Carina M. Schlebusch [1,23,24,25] ✉

Africa harbors the richest diversity of mitochondrial DNA lineages, reflecting its central role in human evolutionary history. Early studies of mtDNA variation provided the first genetic evidence for the African origin of modern humans. With complete mitochondrial genome sequencing, we can now reconstruct maternal lineages with high resolution, yet large parts of the continent remain underrepresented. Using a newly developed long-range sequencing assay, we generated 1176 complete mitochondrial genomes from 13 countries across sub-Saharan Africa, focusing on previously understudied regions. We combined these with over 3600 publicly available African mitogenomes to produce a comprehensive dataset and updated overview of maternal genetic diversity across the continent. We contextualized this diversity with autosomal structure and information on major human expansions, integrating archeological and linguistic evidence. Our analyses suggest an initial demographic expansion of Niger-Congo speakers around 17 thousand years ago (kya), and an initial expansion associated with Bantu-speaking groups around 6 kya. We identify haplogroup L3e as a key marker of this early Bantu expansion, tracking its spread across sub-Saharan Africa. Distinct demographic signatures also emerge for different geographic sub-branches of Bantu speakers. These findings highlight the power of mitochondrial DNA to trace maternal ancestry and demographic history in Africa, while also acknowledging its limitations for phylogeographic reconstruction.

Genetic studies of human populations have described different degrees of structure between and within continents[1–3], with patterns of relatedness often correlating with geography[2,4,5]. This holds for both autosomal variation and for the maternally inherited mitochondrial DNA (mtDNA). MtDNA is often employed for studying population origins, diversity, and migration history because of its high mutation rate, small size (16569 base pairs), uniparental inheritance, lack of recombination, and abundance in cell copies[6]. Given its maternal inheritance, the mtDNA genome's sequence provides unique information on an individual's maternal ancestry. Moreover, the relevance of mtDNA has recently been highlighted by studies of ancient DNA, as it is more easily retrievable in context of damaged and degraded DNA material.

Numerous studies have focused on one or more of the hypervariable regions of the mtDNA genome, sometimes also including haplogroup-defining SNPs[7–11]. Methods to sequence mtDNA have been available since 1977[12] and have constantly become less complex and faster. In 1981, the complete sequence of the human mtDNA genome was published[13]. The first full mtDNA genomes were amplified using PCRs of multiple overlapping fragments, which was recently reduced to amplification in two fragments[14].

Closely related mtDNA sequences are grouped together in mitochondrial haplogroups, collections of similar mtDNA sequences that share single nucleotide polymorphisms (SNPs) inherited from a common ancestor. Mitochondrial haplogroups are conventionally named after capital letters. All modern humans carry mitochondrial haplogroups within

the macrohaplogroup L, which is further subdivided into subclades L0 to L7. The highest diversity of mtDNA sequences is found in Africa, which has led to considerable scientific attention on mtDNA genomes from the continent. Haplogroup L3 gave rise to all mtDNA sequences outside the African continent (belonging to haplogroups M, N and R)[3,15,16]. Although mitochondrial haplogroups M1 and U6 originated outside Africa, they are generally considered African haplogroups, as they were reintroduced by back-migrations and are predominantly found in Africa[17–19].

Linguistic and geographic structure often correlates with genetic structure in Africa[4,5,20,21]. Traditionally, four major indigenous language phyla are identified in Africa:[22] Khoisan, Niger-Congo, Nilo-Saharan, and Afro-Asiatic (Supplementary Fig. 1). Today, the genealogical unity and internal structure of several of these phyla is debated[23] (Figure 1 from Dimmendaal, 2008[24]). However, previous genetic studies used the proposed linguistic phyla to group populations into distinct macroregions which correspond to boundaries of genetic structure, with the suggestion that speakers of related languages would be sharing a demographic history (Supplementary Note 1)[20,25]. In this study, we therefore use similar groupings based on anthropological, linguistic and geographic information to explore demographic history, patterns of genetic structure and signatures of expansion associated with language families.

One early-diverging genetic ancestry is mostly found in people speaking Khoisan languages. Khoisan languages are languages sharing the extensive usage of click sounds. They were once grouped together as one phylum[22], but they comprise at least three distinct families in Southern Africa, as well as two isolate languages in Eastern Africa, i.e., Hadza and Sandawe. The three distinct families in Southern Africa are collectively referred to as Southern African Khoisan (SAK), and these are Kx'a (or Ju, referred to as Northern Khoisan), Tuu (Southern Khoisan), and Khoe-Kwadi (Central Khoisan)[26]. SAK languages count roughly 250,000 speakers today. Throughout this paper, we refer to the languages as *Khoisan* and the people as *Khoe-San* (the designation preferred by the San Council)[27]. The Khoe-San comprise hunter-gatheres (San) and herders (Khoi or Khoe-khoe). The Khoe-San represent one of the two branches in the earliest population divergence among *Homo sapiens*[28–31]. A much more widespread genetic ancestry in sub-Saharan Africa is associated with the Niger-Congo phylum. Niger-Congo is a phylum whose internal classification is debated. Broad agreement exist on a closer relatedness of the Volta-Congo languages[32], which includes the large Bantu family and related languages from Western Africa. There are 400–600 million Niger-Congo speakers[33,34]. According to autosomal microsatellite data, an expansion associated with Niger-Congo speakers started around 7.4 kya[35], whereas the origin of the Niger-Congo dispersal is commonly associated with the African Humid Period at the beginning the Holocene (the current geological epoch that started 11.7 kya)[36,37]. The region of origin of the expansion of Niger-Congo speakers is currently unknown, but North Africa, as well as various regions in West Africa have been proposed[32,38]. Newer linguistic classifications contest the inclusion of the Mande and Ubangian lineages into Niger-Congo[24]. Within Niger-Congo, more than 250 million individuals speak one or more Bantu languages[34]. The vast spread of Bantu speakers across Africa is due to a migration process known as the Bantu Expansion, which started in West Africa (Nigeria/Cameroon) around 5–3 kya[39–43].

Nilo-Saharan languages are spoken in Northeast and Eastern Africa, in the upper parts of the Nile and Chari rivers[44]. There are about 70 million Nilo-Saharan speakers today[33]. Nilo-Saharan's genealogical coherence is contested, but there are some specific genetic ancestries shared among Nilo-Saharan speakers (see Figure 5 and S19 from Tishkoff et al[20]). Alongside Nilo-Saharan-related ancestry in Eastern Africa, Afro-Asiatic-related ancestry is another ancestry found among Eastern African populations. Afro-Asiatic languages are subdivided into six families: Berber, Chadic, Cushitic, Egyptian, Omotic and Semitic[45] and are spoken by some 650 million people today in Northern, Northeastern and Eastern Africa, including the Horn of Africa, and in the Middle East[33]. Indo-European languages like English, Afrikaans, French, Portuguese, and Spanish were also spoken from the 15th century onwards in Africa.

Genetic signatures associated to specific regions and/or language families can be found also in the mitochondrial haplogroups from Africa[1,46–48]. For classification of the geographic areas, consistently used throughout this study, see Supplementary Fig. 2. L0d, one of the two subclades of the most deep-rooted clade of the mtDNA phylogeny[1], occurs primarily among the Khoe-San people in South Africa, Namibia and Botswana[1,8,10,49,50]. Specific subgroups of L0d, namely L0d3, L0d2a and L0d1b, show higher frequencies in the far south, wheareas L0d2c and L0d1a are distributed more centrally within Southern Africa[8]. The majority of L0d subgroups shows significant signs of expansion[8]. Individuals carrying mitochondrial haplogroup L0k generally have a more northern distribution than individuals carrying L0d lineages: L0k is also mostly found among Khoe-San individuals, in Namibia, Angola, Botswana[50] and Zambia[8]. L0a, on the other hand, is very common and widespread. It has been proposed that L0a originated in East Africa[51], but the highest frequencies of L0a are currently found in Mozambique. This distribution has been attributed to the Bantu Expansion[51].

The expansion of Bantu-speaking people is associated with various haplogroups which include L0a[52–54], L1c[55,56], L2a[46,54], L3b[57], and L3e[58,59]. Haplogroup L1 is most frequent in West and Central Africa[60,61], with sublineages L1b frequent in West Africa (Ghana and Ivory Coast)[62], and L1c frequent in Central Africa (Cameroon and Gabon), especially among Western rainforest hunter-gatherers (RHG) (sublineages L1c1a, L1c4 and L1c5)[63–65]. Other sublineages of L1c (L1c1b, L1c1c and L1c2), which are associated with Bantu-speaking people, show signs of recent expansion[63]. L2a is the most frequent mtDNA haplogroup in Africa[62] and is prevalent in many parts of the continent, but is specifically abundant in Central Africa[66]. Haplogroup L3 encompasses African-specific branches as well as non-African branches. L3e is the most widespread branch, and a subclade L3e1 is common among South-Eastern Bantu speakers[67]. L3b and L3d are found mainly in West Africa and L3f is found most frequently in East Africa[46], where it is found in Afro-Asiatic-speaking groups. Its subgroup L3f3 is found in Chadic speakers living today in Lake Chad region[47]. East Africa also hosts the more rare haplogroups L4–L7[60,62]. Haplogroup L0f is characteristic in Afro-Asiatic groups[68]. No mitochondrial haplogroups have been specifically associated with Nilo-Saharan-speaking groups, except a high proportion of L0a (otherwise associated with the Bantu Expansion), which is present among Kenyan Nilo-Saharan speakers[68].

Despite significant advances in our understanding of human maternal history over the past decades, studies on full mtDNA genomes have frequently been limited to specific regions or populations in Africa, or have focused solely on particular mitochondrial haplogroups[17,37,47,48,69–71]. When Behar et al.[72] conducted a reassessment of the mtDNA phylogeny with full genomes, the geographic coverage of the African continent was limited. On the other hand, the last full overview of mtDNA diversity on the African continent performed by Salas et al.[46] has not been updated with full mtDNA genomes. Here, we present 1176 newly sequenced full mtDNA genomes merged with 3612 published full mtDNA genomes from the African continent, serving as a new, up-to-date overview and summary of African maternal history. We outline the phylogenetic relationships between the 4788 mtDNA sequences, examine their contemporary distribution, describe potential regions of origin for the most common mitochondrial haplogroups, and investigate female effective population size changes. By providing a more comprehensive and geographically inclusive analysis, this study offers novel insights into the evolutionary history and demographic patterns of maternal lineages across Africa.

## Results
We first examine the samples generated within this study and then integrate these findings into the larger continental comparative dataset. This approach allows us to compile an inclusive overview of the maternal haplogroups and ancestries across different regions and language groups in Africa, providing insights into the continent's genetic diversity and historical migrations.

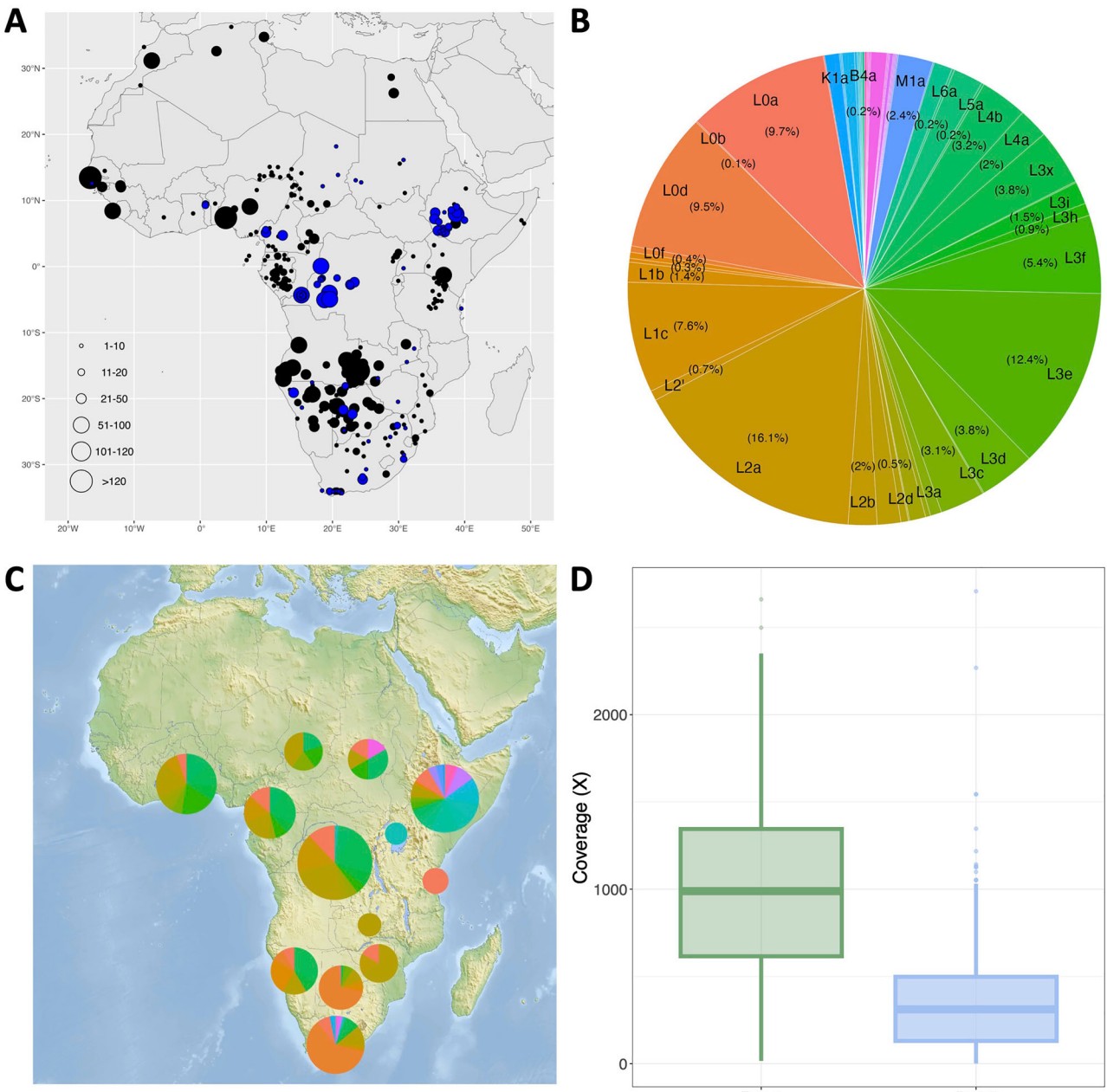

**Fig. 1 | Information on sampling location, sequence coverage, and haplogroups of 1176 newly sequenced individuals. A** Locations of the newly sequenced samples (blue) and the comparative data (black). Size of the circles corresponds to the sample size at that location. **B** The frequency of haplogroups among the newly sequenced individuals. **C** Pie charts showing the frequency of haplogroups among the newly

sequenced individuals per country. Pie charts from countries with a higher number of individuals were depicted larger, colors correspond to those used in **B**. **D** Boxplot of mtDNA coverage for the individuals, separated into the two sequencing runs (n = 292 and n = 1024 individuals respectively).

## Overview of newly generated samples

For this study, we produced complete mtDNA genomes from 1176 new samples from understudied regions within Africa, including Central and East Africa. The individuals come from 71 sites across 13 different African countries (blue circles in Fig. 1A). Full mtDNA sequences were amplified using long-range PCR and sequenced on the PacBio Sequel II platform in two sequencing runs. The average coverage for the first run, encompassing 292 samples, is 994x and the average coverage of the second run including 1024 samples is 340x (Supplementary Fig. 3). Average mtDNA coverage achieved with both PacBio Sequel II sequencing runs surpasses coverages that we achieved previously with the PacBio Sequel (87 samples with an average coverage of 292x)[73], despite the fact that three to twelve times as

many samples were pooled on the PacBio Sequel II. The total number of initial samples (1316) was filtered with a minimum of 30x coverage, which resulted in the exclusion of 130 low-quality samples. In addition, 10 technical duplicates were removed prior to downstream analyses.

Haplogroups were assigned using HaploGrep3[74]. Haplogroup composition (Fig. 1B), shows that L2a (16.1%), L3e (12.4%), L0a (9.7%), L0d (9.5%), and L1c (7.6%) occur most frequently. Mitochondrial haplogroups per site are shown in Supplementary Fig. 4. All haplogroups were associated with a maternal ancestry, according to information available in previous studies (Supplementary Table 1) using language families or geography (Supplementary Fig. 5 by site and Supplementary Fig. 6 by country). We confirm high frequencies of haplogroups L0d and L0k (associated with

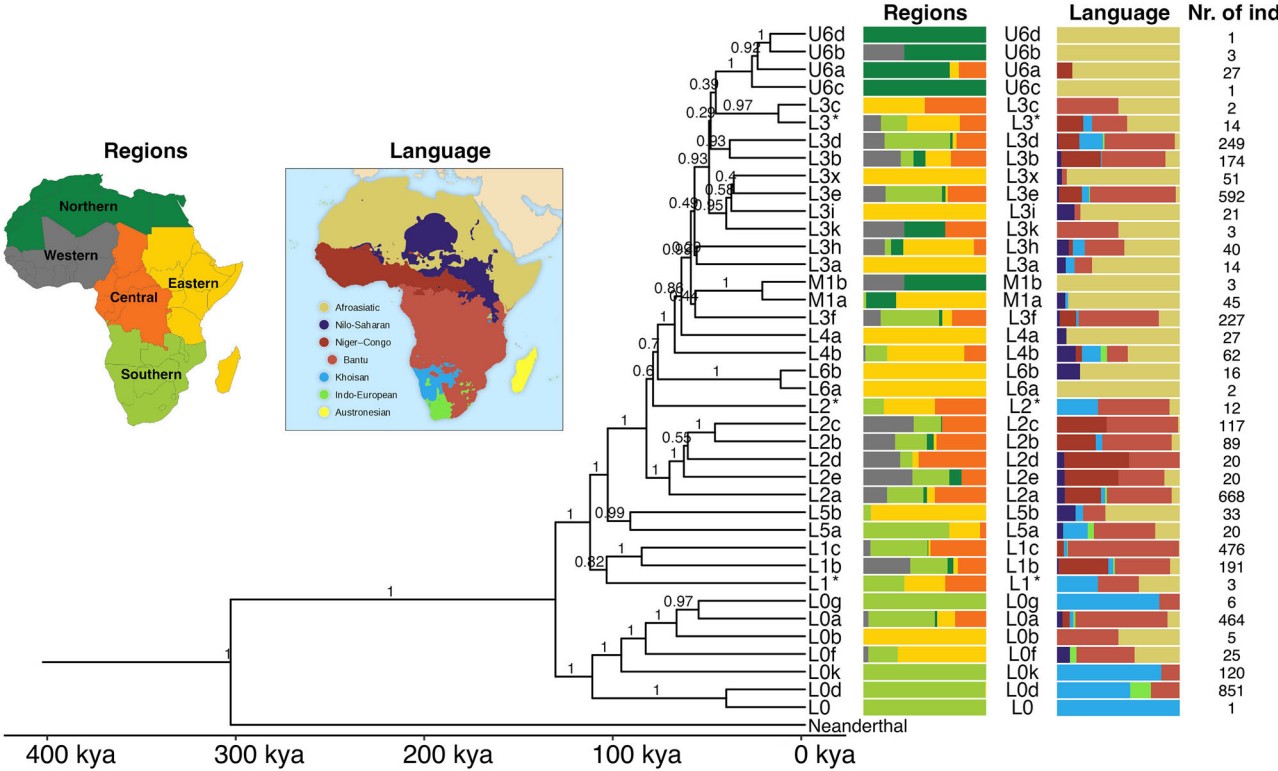

**Fig. 2 | The Bayesian tree topology of African mitochondrial haplogroups based on 39 samples representing major haplogroups.** On the right side of the tree, the regional and linguistic affiliation of the samples belonging to the corresponding leaves are shown. On the right, the number of samples belonging to each leaf. The inserts show the proportion of individuals according to geographic regions or language group. Posterior probability scores are shown at the nodes. Haplogroups denoted with an asterisk (*) contain all sequences not belonging to the other sub-haplogroups of that clade. For example, L1* contains all L1 sequences not belonging to L1b or L1c. The mutation rate used for this figure is $2.285 \times 10^{-8}$ per site per year[60], and TMRCAs calculated using alternative mutation rates are provided in Supplementary Table 2.

Khoe-San ancestry) in Southern Africa (South Africa, Botswana, and Namibia); L0a, L2a, and L3e (associated with the Bantu spread) widespread across sub-Saharan Africa; L3f, L4a and L4b in East Africa (but also in appreciable frequencies in Chad and Cameroon). We note the presence of 58 sequences (5%) with non-African mitochondrial haplogroups (of Asian and European origin). The majority of these (83%) are from Ethiopia.

### Description of database containing nearly 5000 complete mtDNA sequences

A total of 3612 mtDNA sequences from 40 published studies were retrieved from NCBI (Supplementary Data 1). Mitochondrial haplogroups were assigned for all these individuals using HaploGrep3 (Supplementary Fig. 7A). This was combined with the mtDNA sequences of the 1176 newly sequenced individuals to make a total of 4788 mtDNA sequences (Supplementary Fig. 7B). From this point onward, the analyses will focus only on this combined dataset. The most common haplogroup in our African dataset is L0d (17.8%), associated with Khoe-San maternal ancestry, followed by L2a (14.0%) and L3e (12.4%), both associated with the Bantu Expansion.

To provide a general overview of the phylogeny of African mitochondrial haplogroups, we generated a Bayesian phylogenetic tree with a selection of 39 mtDNA sequences representing all major African haplogroups in the dataset up to two classification levels (e.g., L0d, L2a), with a Neanderthal mtDNA as an outgroup, and calibrated with the mutation rate of $2.285 \times 10^{-8}$ per site per year[60] (Fig. 2). The bar charts on the right side of the figure depict the composition of broad linguistic groupings and geographic regions associated with the individuals belonging to each haplogroup tip. We infer the first split in the modern human mtDNA phylogeny to occur between haplogroup L0 and all other mitochondrial haplogroups around 132 kya (95% CI: 117–148). The TMRCA of

individual haplogroups can be found in Supplementary Table 2. We report two different TMRCA estimates: one based on a mutation rate of $2.285 \times 10^{-8}$[60], and one recalibrated from the $2.285 \times 10^{-8}$ rate to account for the effects of purifying selection, following Soares et al.[75]. Although the mitochondrial mutation process is known to be non-linear due to the effects of purifying selection, the Maier et al.[60] rate represents a widely used empirical estimate derived from averaging multiple rates previously applied in the literature. We employ this linear rate as the main rate used throughout the manuscript because it is consistent with the assumptions implemented in BEAST and allows direct comparison with other recent studies using the same rate. The recalibrated values are reported in Supplementary Table 2 and are consistently older, with the largest differences observed for the deepest nodes, such as the TMRCA of all modern humans. When interpreting the uncorrected ages, it should therefore be kept in mind that these likely represent minimum estimates, as the non-linearity of the mitochondrial mutation process was not explicitly modeled in the BEAST analyses.

In the following analyses, when referring to Niger-Congo, we consider only Niger-Congo speakers speaking non-Bantu languages, as well as individuals speaking Mande languages. We note that no individuals speaking Ubangian languages are present in this group, as this linguistic branch is of disputed affiliation to the Niger-Congo phylum. These Niger-Congo groups have a Western African to West-Central African distribution and are concentrated above the equator. Bantu-speakers have a wider distribution across sub-equatorial Africa and are analyzed in a separate analysis. In general, very few haplogroups exhibit a direct one-to-one relationship with either region or language; however, there are specific patterns that emerge and haplogroups that are present in higher frequencies in certain regions or with certain linguistic affiliations. We provide an overview of the connections between different mitochondrial haplogroups

**Fig. 3 | Nucleotide diversity (left) and frequency (right) maps for five most common haplogroups in our dataset (in alphabetical order: L0a, L0d, L1c, L2a and L3e) visualized on a map.** Nucleotide diversity maps are based on 3 degrees bins and a minimum of 10 individuals for nucleotide diversity calculation, frequency maps are based on 1 degree bins and a minimum of 10 individuals.

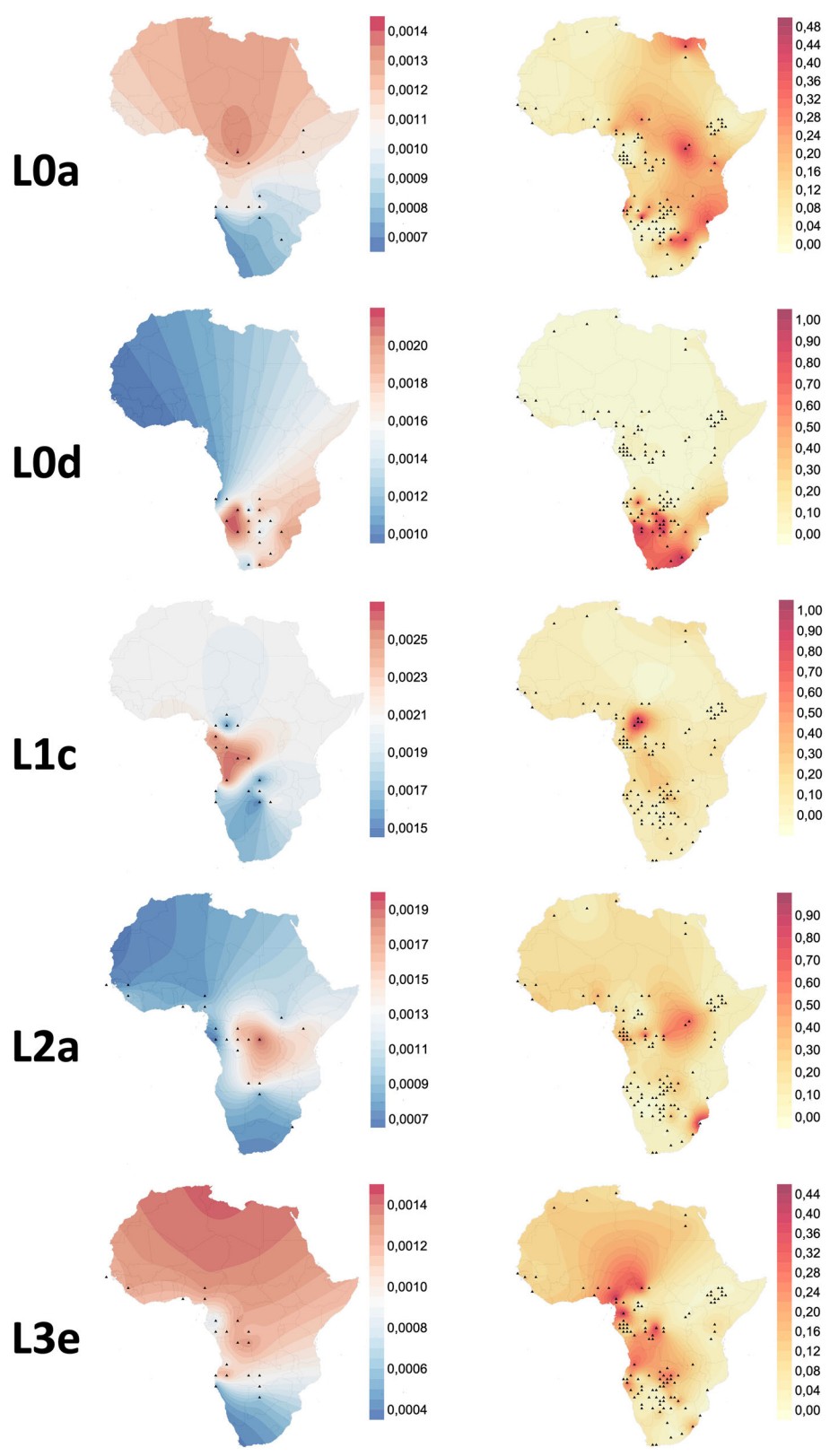

and African regions as well as language groups based on four different analyses:

1. Geographical distribution visualized on a map (Fig. 3 and Supplementary Fig. 8), for the ten most frequent haplogroups in our dataset, showing the current frequency distribution of haplogroups.

2. Nucleotide diversity ($\pi$) visualized on a map (Fig. 3), quantifying the genetic variation among individuals. The nucleotide diversity of individuals carrying a specific haplogroup tends to be elevated in regions associated with the origin of that haplogroup. Thus, when applied to the mtDNA genomes of individuals sharing the same haplogroups, this

analysis provides valuable insights into potential regions of origin for maternal lineages.

3. Phylogenetic relationships between the mtDNA genomes of the individuals (Fig. 2 and Supplementary Figs. 9–16), including information about the estimated Time to the Most Recent Common Ancestor (TMRCA).

4. Analysis of the regional distribution and languages of the individuals within haplogroups (Fig. 2). Hereby, we provide an overview of which languages are spoken by individuals carrying specific mitochondrial haplogroups, as well as in which regions these haplogroups mostly occur.

**Haplogroup descriptions.** The first split in the modern human mtDNA phylogeny is between **L0** and all other mitochondrial haplogroups.

Haplogroup **L0d** is exclusively found in Southern Africa and mostly in Khoisan speakers (59.6%). A smaller proportion of individuals carrying L0d are Bantu speakers (23.3%); i.e., Eastern Bantu (12.4%) and South-Western Bantu (9.9%). All the L0d-carrying Afrikaans speakers (16.9%) are from Namibia[76], or self-identify as Coloured from South Africa[77]. The most recent common ancestor of all L0d sequences lived 95.5 kya (95% CI: (76.0–117.7)). The highest nucleotide diversity is found slightly west of the Kalahari desert (Fig. 3).

**L0k**, with a TMRCA of 64.7 kya (95% CI: 47.5–82.8), has the highest occurrence in northeastern Namibia and northern Botswana, and is less widespread than L0d. The majority of the sequences belong to L0k1 (90%), and more specifically to L0k1a1 (58% of all L0k sequences). This haplogroup starts to diversify around 10–12 kya. Most of the individuals (85%) carrying haplogroup L0k speak Khoisan languages, and the remaining 15% speak Bantu languages, more specifically Eastern Bantu (10.8%) and South-Western Bantu (4.2%).

Haplogroup **L0f** is a rare haplogroup found in Eastern Africa. All L0f sequences share a common matrilineal ancestor around 81.7 kya (95% CI: 70.4–93.3) and diverges roughly 75–50 kya, earlier than L0k. Bantu speakers (47.4%) and Afro-Asiatic speakers (36.8%), make up the majority of the L0f carriers. Eastern Africa hosts mainly L0f2 (connected to Afro-Asiatic speaking individuals), whereas Southern Africa mainly hosts L0f1 (connected to Bantu speaking individuals). The single individual with haplogroup L0f from West Africa carries L0f2a.

Within L0, **L0a** (TMRCA of 73.2 (95% CI: 61.1–86.0)) has a peculiar geographic distribution, different from the sister branches L0d, L0k and L0f. L0a is very widespread, and is found at frequencies up to 40% in the northeastern part of the Democratic Republic of the Congo (DRC), the northeastern part of South Africa, and even in northern Egypt. It is more or less absent in the very southwestern part of the African continent. Although most of the L0a carriers are Bantu speakers (79.8%), it can be found among all language groups. The highest nucleotide diversity is found in the DRC and South Africa. The remaining L0 branches **L0b** and **L0g** appear at a very low frequency, in East and Southern Africa respectively.

Following L0, haplogroup **L1** is the next haplogroup to split off from all the other haplogroups around 105 kya (95% CI: 87.8–122.5), with its subclades L1b and L1c sharing a common ancestor 86.7 kya (95% CI: 69.8–102.9). Haplogroup **L1b** is found mainly in West (38.2%) and Southern Africa (30.4%). L1b carriers mainly speak non-Bantu Niger-Congo and Mande languages (51.2%), and Bantu languages (36.3%), and to a small extent Khoisan languages (3.2%). Haplogroup **L1c** is among the most frequent haplogroups in our dataset, with highest frequencies in the homeland of the Bantu Expansion (Nigeria and Cameroon). It is also found in the DRC, Angola, Zambia, and Namibia. The highest nucleotide diversity is in the western part of Central Africa. L1c carriers mainly speak Bantu languages (88.8%), and to a smaller extent Khoisan and non-Bantu Niger-Congo languages, including Mande (2–8%). The TMRCA of L1c sequences is 83.9 kya (95% CI: 73.5–95.2).

Mitochondrial haplogroup **L2** splits off after haplogroup L5 (L5 discussed below) 84.1 kya (95% CI: 74.8–94.0). Interestingly, all L2 haplogroups (L2a–L2e) are characterized by linguistic and regional

heterogeneity, with a broad range of linguistic affiliations and geographic coverage. Specifically, there are prominent proportions of Bantu (33.3-50.6%) and non-Bantu Niger-Congo and Mande speakers (34.9-61.7%). Moreover, the highest frequencies are found in Central, Western, and Southern Africa. A small proportion of L2a and L2b carriers are Khoisan speakers (3.0 and 4.5% respectively). The majority of L2 individuals in our dataset belong to haplogroup **L2a** (73.0%), with a TMRCA of 58.4 kya (95% CI: 48.9–68.4). The highest nucleotide diversity of L2a is in Central Africa. It is prevalent in many parts of the African continent, but the highest frequencies are found in northeastern DRC and Mozambique. 48.8% of people belonging to haplogroup L2a speak Bantu languages, 34.9% speak non-Bantu Niger-Congo and Mande languages.

Haplogroup **L3** has an extensive global distribution; all haplogroups outside of Africa trace back to the ancestral lineage of haplogroup L3. Thus, it plays an important role in understanding the out-of-Africa expansion. Additionally, L3 haplogroups have a wide distribution across the African continent, with presence in all five geographic regions and among all language groups. Contrary to what their names might imply, M1 and U6 are subgroups of L3. Haplogroup **L3b** is found in 3.6% of our dataset, with highest frequencies in East Africa, in Kenya specifically. Interestingly, it is found in relatively equal proportions in all five African regions, ranging from 9.7% in North Africa to 30.3% in West Africa. Both Bantu speakers and non-Bantu Niger-Congo and Mande speakers make up slightly more than 40% of the individuals carrying haplogroup L3b. Haplogroup **L3d** occurs at a frequency of 5.2%. 53.4% of the L3d sequences are from Southern Africa. 52.2% of L3d individuals speak Bantu languages, 24.7% speak non-Bantu Niger-Congo and Mande languages, and 17.4% Khoisan languages. Haplogroup **L3f** shows a similar distribution pattern to L3d. It is found in 4.7% of the dataset with highest frequencies in northern Namibia. Significantly, 65.2% of L3f individuals speak Bantu languages, and 47.4% live in Southern Africa. **L3e** frequency in the dataset is 12.4%. The highest frequencies are widely distributed from the homeland of the Bantu Expansion to northern Namibia. L3e is mainly found in Southern Africa (45.9%), Central Africa (31.0%), and West Africa (18.2%). The highest nucleotide diversity is in Western and Central Africa. The TMRCA of all L3e sequences is 36.1 kya (95% CI: 30.0–42.4). The majority of L3e carriers speak Bantu languages (67.0%). For haplogroup L3e1 and L3e2, we generated median joining networks (Supplementary Figs. 17 and 18) to visualize the genetic relationships between haplotypes and infer evolutionary connections. For haplogroup L3e1, individuals with southern origins are located more at the edges of the network, whereas individuals with Central and Western origins are located more centrally in the network. This pattern is not observed for L3e2.

Haplogroups **L4, L5, and L6** are mostly found in Eastern Africa. Haplogroups L4a, L6a, and L6b are exclusively found in East Africa and almost exclusively among Afro-Asiatic speakers. L5a has a more southern distribution, with 66.7% of all our L5a sequences coming from Southern Africa. Further, 50.0% of the L5a carriers speak Bantu languages.

Although mitochondrial haplogroups **M1 and U6** originated outside Africa, they are generally considered African haplogroups, as they were reintroduced by back-migrations and are now predominantly found in Africa[17–19]. Haplogroups U6a, U6b, U6c, and U6d are almost exclusively found among Afro-Asiatic speakers and 66.7–100% of the people carrying these haplogroups live in North Africa. Interestingly, 33.3% of the U6b carriers live in West Africa. Haplogroup M1 shows a slightly more diverse pattern for both regions and languages when compared to U6 haplogroups. M1a, which is found in 45 samples in our dataset, occurs mostly in East Africa (72.3%) and 90.9% of M1a carriers are Afro-Asiatic speakers. These haplogroups are represented by relatively few individuals in our dataset, and more sequences would be needed to consolidate these haplogroups profiles.

## Variation in female effective population sizes associated to language groups

All individuals were assigned a language group if language information was available. We refer to Supplementary Note 1 for the use of linguistic labels

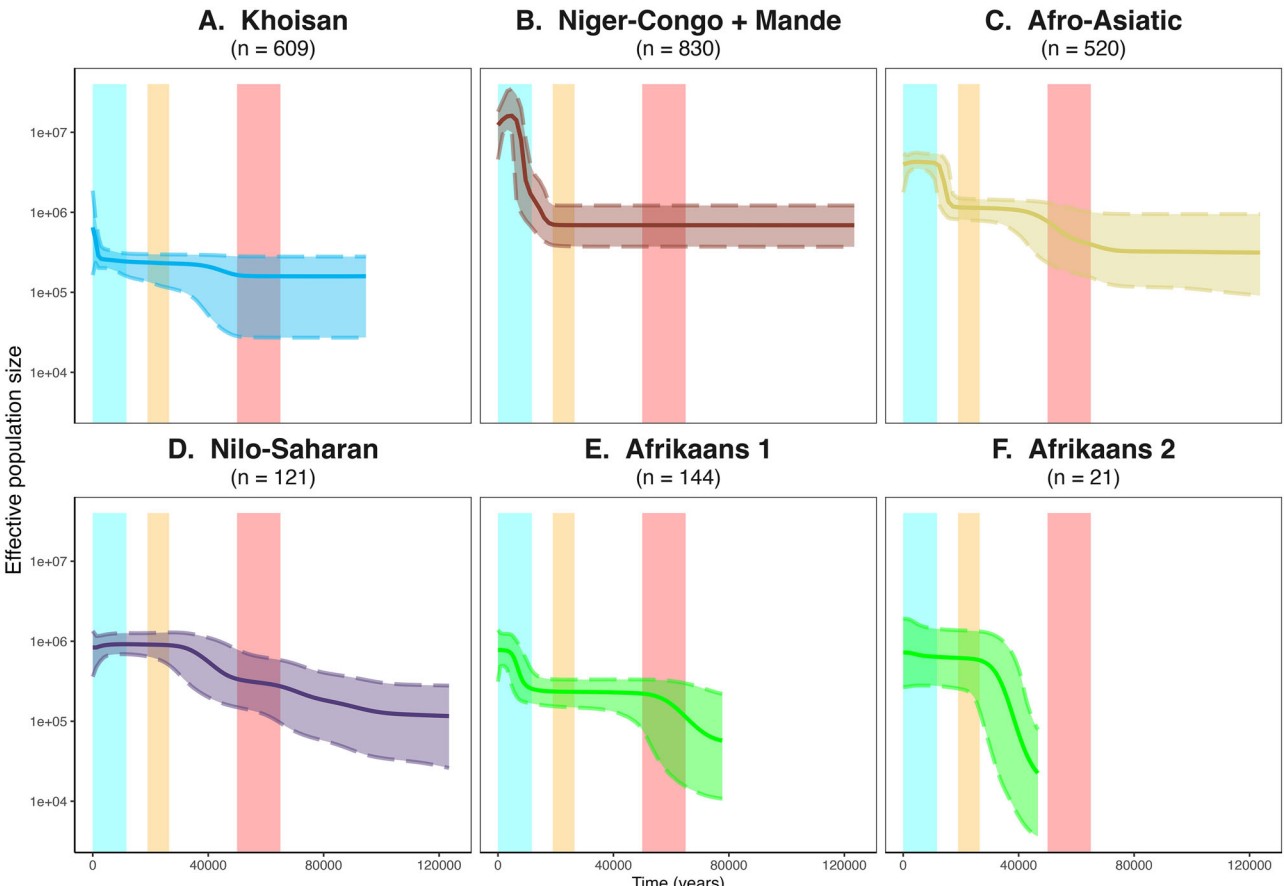

**Fig. 4 | Bayesian Skyline Plots (BSP) showing variation of $N_e$ through time for separate language groups.** **A** Khoisan (only L0d and L0k carriers), **B** Niger-Congo and Mande (excluding Bantu speakers), **C** Afro-Asiatic, **D** Nilo-Saharan, **E** Afrikaans 1 (including only L0d and L0k haplogroups) and **F**) Afrikaans 2 (including only haplogroups of Eurasian origin) speakers. The bold middle line represents the mean estimates and the two dashed lines represent the 95% highest posterior density (HPD) intervals. The number of individuals analyzed in each plot is indicated in parentheses. The red area denotes the time of the out-of-Africa migration (65--50 kya), the yellow area the Last Glacial Maximum (LGM)(26.5--19 kya), and the blue the Holocene (last 11.7 ky). $N_e$ is represented with a log-scale on the Y-axis, and the years in the past are represented on the X-axis. All sub-plots share the same axes. A zoom-in of the last 25 000 years for Niger-Congo and Mande can be found in Supplementary Fig. 22.

for genetic clusters. Comparisons were made between Afro-Asiatic, Nilo-Saharan, Bantu, non-Bantu Niger-Congo and Mande, Khoisan, and Afrikaans (an Indo-European language of Dutch origin that developed locally). The distribution of mitochondrial haplogroups among these different groups (Supplementary Fig. 19) illustrates the maternal lineage diversity within the African continent. Variation of female effective population sizes ($N_e$) through time for each of these groups was determined through Bayesian Skyline Plots (BSP) (Fig. 4). $N_e$ of various Bantu speaking groups (North-Western Bantu, West-Western Bantu, South-Western Bantu and Eastern Bantu) were analyzed separately and are shown in Fig. 5.

**Khoisan.** Because genetic studies have been focusing intensively on Khoe-San populations due to their importance for deep population history inference and distinct uniparental lineages, the dataset includes a relatively large number of Khoisan speakers (777 individuals, roughly 16.2% of the dataset). The predominant mitochondrial haplogroup among people speaking Khoisan languages is L0d (65.3%). Khoisan speakers show a slight increase in female $N_e$ after 65–50 kya, which interestingly corresponds to the time period just after the out-of-Africa expansion (Fig. 4A). Fig. 4A is based only on L0d and L0k haplogroups, as these were the only haplogroups found among Southern African Stone Age hunter-gatherers before Bantu speakers and East Africans speaking other languages moved into the area, as shown by aDNA studies[30,78]. For a complete analysis of the effective population size of all haplogroups currently found among Khoisan speakers see Supplementary Fig. 20. We

also characterized the haplogroup distribution among six different Khoisan groups (Kalahari Khoe, Khoekhoe, Kx'a, Tuu, Sandawe and Hadza)(Supplementary Fig. 21), and observe different haplogroups among the Sandawe and Hadza, although only a few individuals from these groups were included in the dataset. Furthermore, it seems like Khoekhoe and Kalahari Khoe have more external input (non-L0d/k) compared to Tuu and Kx'a. It is however difficult to deduce whether this input is from Bantu speakers or East African groups speaking other languages.

**Niger-Congo including Mande.** Individuals speaking non-Bantu Niger-Congo and Mande languages are analyzed together as they share a similar autosomal genetic ancestry[20]. In our analysis, we group individuals spanning a region from Senegal in the West to the Central African Republic in the East. Major haplogroups found among these individuals are L2a (27.8%), L3e (15.3%) and L1b (11.9%). In this group, we see a population expansion starting around 17 kya, with a maximum growth around 9 kya (Fig. 4B and Supplementary Fig. 22). Moreover, there seems to be a decrease in $N_e$ from roughly 4 kya. We also analyzed the female $N_e$ of Niger-Congo and Mande speakers including Bantu speakers (Supplementary Fig. 23), which shows two expansion periods, one starting around 19 kya, and the other starting around 5 kya.

**Afro-Asiatic.** Afro-Asiatic speakers in our dataset mainly come from the northern part of the African continent. Although the database selection

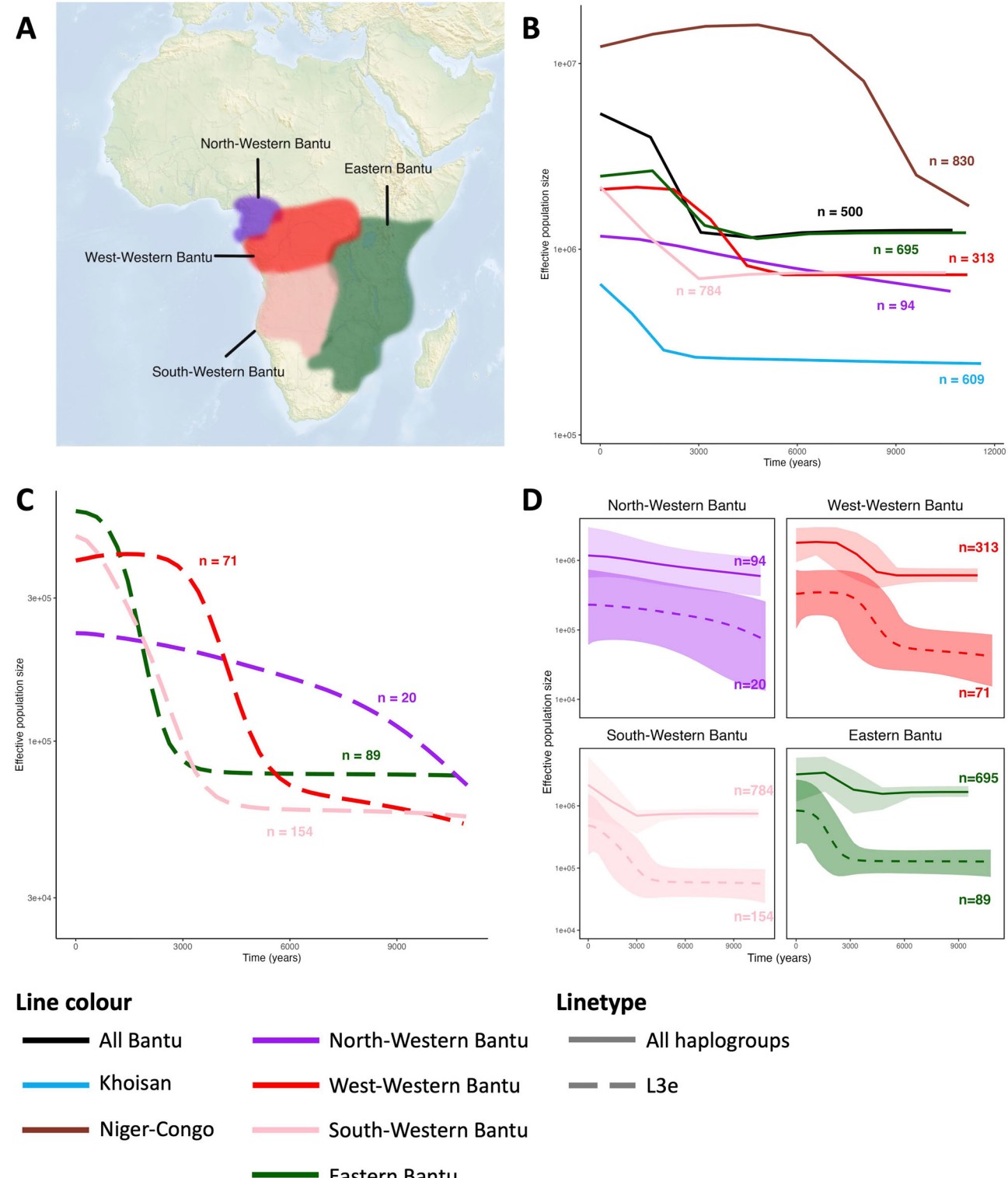

**Fig. 5 | Variation of female $N_e$ through time for separate Bantu speaking groups.** **A** The approximate geographic distribution of four sub branches of Bantu languages. **B** $N_e$ variation for the different Bantu speaking groups. Detailed figure including confidence intervals can be found in Supplementary Fig. 25. **C** $N_e$ variation from individuals carrying haplogroup L3e in the various Bantu speaking groups. **D** $N_e$ variation from 1) all individuals within a language group (dotted line) and 2) only individuals carrying L3e within that language group. The bold middle line represents the mean estimates and the faded area surrounding the bold line represent the 95% highest posterior density (HPD) intervals. Effective population sizes are represented on a log-scale on the Y-axis, and the years in the past are represented on the X-axis. All sub-plots in D) share the same axes. The number of individuals used for each BSP is indicated next to the line in the corresponding color.

https://doi.org/10.1038/s42003-026-10330-9                                                                                    **Article**

was restricted to African haplogroups, non-African haplogroups of European origin were found among the 1176 newly sequenced mitogenomes: like R0a at 1%, K1a at 1%. The Afro-Asiatic group includes haplogroups that originated outside of Africa and returned through back-migration (M1 at 8.3% and U6 at 5.0%), as well as haplogroups associated with Western African non-Bantu Niger-Congo and Mande, and Bantu speakers (L2a (7.7%) and L0a (8.5%)). The BSP shows an increase (three- to ten-fold) after the out-of-Africa expansion, and a subsequent increase starting around 20 kya.

**Nilo-Saharan**. The small number of individuals associated to the Nilo-Saharan group in our dataset (N=121) harbor a wide diversity of mitochondrial haplogroups, like L2a (32.2%), L0a (16.5%) and L3e (7.4%). Individuals associated to the Nilo-Saharan group show an increase of $N_e$ (three- to ten-fold) after the time period of the out-of-Africa expansion, similar to Afro-Asiatic speakers but slightly more delayed. With a relatively small sample size we do not recognize any distinct mitochondrial haplogroup associated to this group (Fig. 2). This reiterates the lack of linguistic and genetic unity within the Nilo-Saharan phylum.

**Afrikaans**. Afrikaans speakers in this study are only individuals self-identifying as "Baster"[76] and "Coloured"[77]. Many first language speakers of Afrikaans in Southern Africa carry the Khoe-San associated haplogroup L0d. The individuals who speak Afrikaans and carry L0d haplogroups show two $N_e$ expansions in the BSP (Fig. 4E); one preceding the time period of the out-of-Africa (65–50 kya), the other starting around 10 kya, which is similar but slightly later than that observed in Afro-Asiatic speakers. Haplogroups that originated in Eurasia, such as M and B are found in low percentages (5.1% and 2.0% respectively) among the sampled Afrikaans speakers. The $N_e$ changes of Afrikaans speakers carrying Eurasian haplogroups can be found in Fig. 4F and shows one expansion from roughly 45 kya until the Last Glacial Maximum (26.5–19 kya). The BSP plot with all Afrikaans speakers can be found in Supplementary Fig. 24 and is similar to that of Afrikaans speakers that carry L0d haplogroups (Fig. 4E).

**Bantu**. The largest linguistic group in our dataset consists of people speaking Bantu languages. This group comprises a large variety of mitochondrial haplogroups, with L1c (18.0%), L3e (17.2%), L0a (14.8%), and L2a (14.4%) showing the highest occurrences, followed by the Khoe-San characteristic haplogroup L0d (8.8%). The BSP for all Bantu individuals is shown in Fig. 5B. An expansion (three-fold increase) starts around 3 kya and peaks around 1.8 kya.

The $N_e$ variation was also analyzed separately for speakers of four Bantu linguistic sub-groups (North-Western Bantu, West-Western Bantu, South-Western Bantu and Eastern Bantu) (Fig. 5B). The patterns of $N_e$ variation differ across these groups. Starting in and around the homeland of the Bantu Expansion, the North-Western Bantu speakers show the earliest signs of expansion: a relatively slow but steady three-fold increase over the last 12 ky. The West-Western Bantu increase begins right after 6 kya and accelerates around 4.5 kya, with stabilization by about 2.2 kya. The Eastern Bantu speakers begin to expand around 4.8 kya and stabilizes by about 1.7 kya. A delayed increase in effective population size can be seen among the South-Western Bantu speakers, which starts around 3 kya and has not stabilized yet. Increases in $N_e$ indicate that the Bantu speakers either picked up more mitochondrial haplogroups while they expanded, or their $N_e$ increased through population expansion, or both.

Within various Bantu language groups, we analyzed $N_e$ variation separately for the L3e individuals (Fig. 5C), as the nucleotide diversity and current frequency of L3e carriers suggests that this haplogroup is associated with the earliest expansions of Bantu speakers (Fig. 3). Distinct patterns can be observed over the last 6 kya for the West-Western Bantu, South-Western Bantu, and Eastern Bantu, at which point they begin to converge backwards in time. In Fig. 5D, we illustrate $N_e$ variation comparing L3e against all the haplogroups of the four Bantu subgroups. Notably, the results seem to indicate that for North-Western, West-Western, and South-Western Bantu groups, their population expansion may have occured after the one reconstructed with the L3e individuals only. Eastern Bantu speakers, however, may present an exception, as their expansion peak appears to predate the one found in L3e. Nevertheless, the large and overlapping credible intervals warrant caution in interpreting these temporal differences.

The same analyses were carried out for haplogroups L1c, L0a, L2a, and L3b (Supplementary Figs. 26–29). $N_e$ variation of L2a within the various language groups seems to show more recent expansions than all speakers of that language (Supplementary Fig. 28), indicating that this haplogroup could have been picked up by expanding Bantu-speaking populations. Higher nucleotide diversity values east of the Bantu homeland support this hypothesis. However, the broad and overlapping credible intervals in the BSP suggest that these temporal differences should be interpreted with caution. $N_e$ variation of L1c carriers (Supplementary Fig. 26) shows a slight indication of expansions in the West-Western Bantu speakers and a slight indication of bottlenecks in Bantu speakers living further away from the homeland (South-Western and Eastern Bantu speakers), although confidence intervals are high. Supplementary Fig. 27 shows similar $N_e$ variation for L0a to the rest of the individuals for each language subgroup.

## Discussion

The African continent is home to vast genetic diversity, which includes mtDNA genomes. As mtDNA is passed on through the female line, the genetic diversity in this part of the genome is informative about the female history of a population. In this study, we report a new assay-design for retrieving full mtDNA sequences using long-read sequencing. We generate novel data from understudied regions such as the DRC and Ethiopia, which is added to a newly generated database of full mtDNA sequences, allowing us to give an overview of the mtDNA diversity of the African continent. We describe different distribution patterns and reconstruct the history of the most frequently occurring mitochondrial haplogroups through frequency maps, nucleotide diversity maps, analysis of $N_e$ variation, and time-calibrated phylogenetic trees. From the combination of these results several important findings about African maternal history came to light.

In a previous study, we amplified and sequenced the mtDNA genome in two fragments[14]. Here, we present a method to sequence the full mtDNA genome with the amplification of a single fragment. It requires fewer consumables, fewer reagents, less time and greater monetary efficiency compared to all other methods to sequence the full mitochondrial genome. Additionally, we show that it is possible to sequence at least 1024 individuals in one run, with a median coverage of 340x (Supplementary Fig. 3). By sequencing 1176 new mtDNA sequences and compiling a database containing nearly 5000 full mtDNA sequences, this study provides the first large overview of mtDNA diversity on the African continent since 2002[46]. We have used published data to associate mitochondrial haplogroups with maternal ancestries described by geography, culture and language. The 1176 new mtDNA sequences, which include underrepresented regions of Central and East Africa, confirm the generic haplogroup structure and the affiliations with linguistic groups or geographical regions proposed (Supplementary Fig. 6).

In this study, we generated data from 13 different countries across sub-Saharan Africa, with a significant number of samples originating from the Democratic Republic of the Congo (DRC, $N = 520$) and Ethiopia ($N = 335$). These regions have been historically understudied, and our research sheds light, for the first time, on the mtDNA diversity of the populations in these countries. In the DRC, individuals predominantly carry haplogroups commonly associated with Bantu-speaking populations and West-Central Africa, including L0a (12.5%), L1c (14.2%), L2a (23.1%), and L3e (21.0%). In contrast, in Ethiopia the East African haplogroups are more frequent, such as L3x (13.4%), L4a (7.2%), L4b (8.4%), L5b (7.2%), and M1a (8.4%). L3x is an otherwise rare haplogroup; the highest frequency was previously detected in Ethiopia at 6% in a Cushitic group[79].

It should be noted that the frequencies of the various haplogroups in our dataset are influenced by sampling bias. As an example, the most

frequent haplogroup in the compiled dataset is L0d (17.8%). However, L0d is common among Khoe-San people, which are known to have a census size much smaller than Bantu speakers[80]. Thus, the high percentage of L0d mtDNA sequences in our dataset is most likely a result of the interest in the Khoe-San people because of their unique placement in the phylogeny of all modern humans. In fact, 18.6% of the individuals in the dataset speak Khoisan languages.

The majority of Khoisan speakers carry haplogroup L0d and L0k. Haplogroup L0d reaches a maximum frequency of 100% at some sites, whereas L0k frequency is much lower, maximum 33%. The distribution of L0k is also more geographically confined than L0d, which has previously been shown by Barbieri et al.[50] However, Barbieri et al. observed peaks of L0k in northern Zambia, which we do not detect, probably due to a more extensive dataset used here. Afro-Asiatic speakers have been associated with haplogroups L0f and L3f[47,68]. We show that these haplogroups do occur among Afro-Asiatic speakers in our dataset, but in low frequencies (1.3% and 7.3%, respectively). Haplogroup L3f additionally shows high frequencies among Bantu-speaking populations in northern Namibia. The haplogroups L0a and L2a, which are common among Bantu speaking populations, also occur at lower frequencies among Afro-Asiatic speakers. Nilo-Saharan speakers, on the other hand, show higher frequencies of some of the haplogroups also associated with Bantu speakers; L2a, L0a and L3e (totaling to 56.1%). L0a and L3e have broad distributions across the African continent, whereas L2a distribution is slightly more restricted. Mande speakers and Niger-Congo speakers whose languages are not Bantu show relatively high proportions of L2a (27.8%), L3e (15.3%) and L1b (11.9%).

Individuals self-identifying as "Baster" and "Coloured", speaking Afrikaans (an Indo-European language) from South Africa, are characterized by a high proportion of the Khoe-San haplogroup L0d (73.1%). This emphasizes the impact of historical events such as colonial movements, migration, and intermarriage on the complex genetic landscape of the region[73,77].

We also note the presence of non-African mitochondrial haplogroups (Asian and European) in various African countries, which can be attributed to either back-migration in the cases of Ethiopia and Sudan, or recent colonial influence in the cases of South Africa, Namibia and Botswana.

The expansion of Bantu-speaking people was one of the most significant population movements in African history and had profound impacts on the demographic, linguistic, and cultural landscape of the continent[5]. Studies based on linguistic evidence have proposed an expansion onset around 5 kya, or slightly earlier[81,82]. The start of this expansion has been dated with autosomal microsatellite data to 5.6 kya[35]. Further studies investigated population expansions within distinct Bantu speaking groups using admixture dating methods and IBDNe, a method to estimate $N_e$ from the number of shared ancestors with identity-by-descent (IBD) segments[5,83–85]. However, IBDNe does not reliably reconstruct relationships over 50 generations ( ~ 1.5 ky)[86]. We have reinvestigated the timing and magnitude of the expansion using a Bayesian approach on mtDNA genomes. The expansion starts earlier for Western Bantu speakers (6 kya), with a slow but steady three-fold increase and starts later for Eastern and South-Western Bantu speakers (4.5–3 kya). This early expansion predates the estimates from microsatellites by 400 years, and those from linguistic evidence by 1000 years. However, it is important to note that this estimate of 6 kya heavily relies on the mutation rate used ($2.285 \times 10^{-8}$ per site per year)[60], which is based on an average of mtDNA mutation rates surveyed in the literature. The ranges found in the literature span from $1.665 \times 10^{-8}$[75] to $2.67 \times 10^{-8}$[87]. Consequently, when calculating this through to a range in year estimates, it comes down to roughly 8.25–5.15 kya.

It is also interesting to investigate the specific genetic signals associated with the Bantu Expansion. The identification of haplogroups linked to the initial dispersal of Bantu speaking peoples remains a subject of ongoing investigation. We propose that lineages within haplogroup L3e were among those associated with the earliest expansions of Bantu speakers. The nucleotide diversity observed among L3e carriers shows a north-south cline, with higher nucleotide diversity in West-Central Africa (Fig. 3). Current L3e

frequencies are high in West-Central Africa, and in regions where Western Bantu speakers live. Within the Bantu speaking individuals carrying haplogroup L3e, separate analyses of $N_e$ variation were conducted for each of the four subbranches of the Bantu language tree (Fig. 5C and D). The BSPs of L3e lineages among West-Western, South-Western and Eastern Bantu speakers exhibit distinct patterns over the last 6 kya, at which point they begin to converge backwards in time. This convergence suggests a scenario wherein these L3e lineages might have been contained within the initial Bantu source population until 6 kya, potentially among the people residing in the Bantu Expansion homeland. However, given that haplogroup L3e as a whole is much older (estimated TMRCA is 36.1 kya, 95% CI: 30.0–42.4), it is unlikely that all L3e lineages are involved in the Bantu expansion. It is therefore more plausible that only particular subhaplogroups of L3e were carried by the first Bantu-speaking groups leaving the homeland. Identifying which subhaplogroups were associated with the expansion will require further research. Moreover, it is important to be careful with the interpretation of $N_e$ variation through time with regards to specific mitochondrial haplogroups. Mitochondrial haplogroups do not represent populations and could have entered a population at any time-point in the past. The BSP of a haplogroup before entering the studied population therefore provides no information on the female $N_e$ of that population. Moreover, the more individuals carrying a specific haplogroup within a language-group, the more the BSP of that language-group will resemble that of the haplogroup.

We also generated median joining networks for haplogroup L3e1 and L3e2 to visualize the genetic relationships between haplotypes and infer evolutionary connections (Supplementary Figs. 17 and 18). For haplogroup L3e1, individuals with Southern African origins appear more at the edges of the network, especially within subhaplogroups L3e1d, L3e1e, and L3e1a2. In contrast, individuals with Central African origins, such as those from the DRC and Cameroon, are more commonly found at the network edges in subhaplogroups L3e1a1 and L3e1a3a. This pattern suggests that these subhaplogroups likely reflect signals of back-migrations, but only for specific clades. For example, extensive back migrations of Bantu-speakers from the South to the North have occurred during the Mfekane migrations associated with unrest and displacement during the rise of the Zulu civilization[88]. Networks do have certain limitations that should be kept in mind, such as the possibility that present-day sequences may occupy internal nodes and that demographic processes are not explicitly accounted for.

The variation in $N_e$ was also investigated for separate haplogroups associated to Bantu speakers (L0a, L1c, L2a, L3b), within the four Bantu subgroups (Supplementary Figs. 26–29). L2a shows expansions in the separate Bantu subgroups that appear more recent relative to the Bantu metapopulation (Supplementary Fig. 28). This pattern could be consistent with L2a having been picked up by expanding Bantu-speaking populations after the start of the Bantu migration, with the lineage itself expanding later within the Bantu population, or with a secondary expansion wave. However, given the broad and overlapping credible intervals in the BSPs, these scenarios remain tentative and should be interpreted with caution. The onset of the $N_e$ expansion in L0a, L1c and L3b carriers occurs at the same time in the four Bantu subgroups and in the general Bantu metapopulation. Expansion signals associated to haplogroups L2a and L3e seem to appear later among Eastern Bantu speakers compared to the overall population of individuals speaking Eastern Bantu languages. This phenomenon is more pronounced in L3e than for L2a.

By comparing the distribution of haplogroup frequencies and nucleotide diversity, we tentatively search for signatures of shifts in spatial distribution (Fig. 3). For haplogroup L0a, the region with highest level of nucleotide diversity (western DRC) does not correspond to the regions with highest frequencies (eastern DRC, Mozambique, North-East South Africa, and Egypt), suggesting a shift eastwards. Even though L0a was previously proposed to have an Eastern African origin[51], the nucleotide diversity and frequency distribution (Fig. 3) here point towards a Central African origin, which is more in line with a Bantu origin - this haplogroup is also found at

high frequency among Bantu-speakers. For L0d, there is high spatial overlap between the highest nucleotide diversity and the highest frequency. This could be explained by the long geographic continuity and isolation of Southern African Khoe-San groups[89].

L1c (Fig. 3) is frequent in the homeland of the Bantu Expansion[64] but the nucleotide diversity is highest toward the south (Angola and southern DRC). Inversely, where L1c shows high frequencies the nucleotide diversity is lowest. We hypothesize that this haplogroup, of which subhaplogroups are present in high frequencies in RHG populations, was more widespread in the past among RHG, and that nucleotide diversity has been reduced through drift and isolation there. Haplogroup L2a has highest nucleotide diversity in the center of the DRC, but highest frequencies in eastern DRC and Mozambique. It is possible that the L2a haplogroup was picked up by the Bantu Expansion and spread eastwards with subsequent local expansions.

We acknowledge the limitations of utilizing nucleotide diversity calculated in present-day human populations as a metric for tracing the origin of a maternal lineage. In instances of complete population replacement in the region of origin, no original nucleotide diversity information can be found, which may result in the identification of other regions as the potential origin for that haplogroup. Moreover, the extrapolation of nucleotide diversity in regions without individuals with that haplogroup using the kriging method should be interpreted with caution.

The expansion of haplogroups associated to non-Bantu Niger-Congo and Mande languages (Fig. 4 and Supplementary Fig. 22) starts around 17 kya and experiences maximum growth around 9 kya, predating previous estimations based on microsatellite data (7.4 kya)[35] and a subset of mtDNA genomes from populations in Burkina Faso (10–12kya)[37]. Although the expansion of Niger-Congo speakers is commonly connected to the stabilizing climate during the Holocene (the current geological epoch that started 11.7 kya)[36], this signal is not replicated with our data. Instead, we see a stronger impact of the end of the Last Glacial Maximum (LGM) (26.5–19 kya), in driving a demographic expansion. This signal of expansion is very deep in time and it is difficult to associate it with the origin of the language family. Linguistic reconstructions beyond 10 kya are generally not easy to infer. This deep expansion connected to a similar genetic ancestry for the region[20] could provide a common demographic substrate for linguistic subbranches (e.g., Mande) whose languages have weaker genealogical connections to the rest of Niger-Congo family.

Moreover, there seems to be a decrease in $N_e$ from roughly 4 kya. This population decline should, however, be regarded with caution, as it is most likely an artefact of population structure rather than a true demographic contraction[90].

## Conclusion

This study provides the most comprehensive overview to date of mitochondrial DNA diversity across sub-Saharan Africa, combining 1176 newly sequenced mitogenomes with over 3600 publicly available sequences. By targeting understudied regions and integrating linguistic, geographic, and archeological data, we uncover complex patterns of maternal ancestry and demographic history, including refined timings for the expansions of Niger-Congo and Bantu-speaking populations. Haplogroup L3e emerges as a key maternal marker of early Bantu dispersals, and regional differences in demographic trajectories among Bantu subgroups reveal the layered nature of population movement across the continent. Importantly, the extensive and geographically inclusive mitogenome dataset generated here serves as a valuable reference for future genetic, archeological, and anthropological research. It offers a framework for comparative analyses, facilitates the interpretation of ancient DNA, and enhances our understanding of human population structure and migration in Africa. While mitochondrial DNA has limitations, its strength in capturing maternal lineages makes it a powerful tool, especially when supported by a reference database of this scale and resolution. This reference dataset will aid future efforts to infer African demographic history, inform ancient DNA interpretations, and improve the resolution of maternal lineage studies on a continental scale.

## Methods

### Sampling and long-read sequencing

Saliva samples or already extracted DNA samples were obtained for 1316 African individuals from 71 different sites spread over 13 different African countries (Botswana, Cameroon, Chad, DRC, Ethiopia, Namibia, South Africa, Sudan, Togo, Uganda, Zambia, Zanzibar, and Zimbabwe). Participants donated saliva samples with written informed consent. All ethical regulations relevant to human research participants were followed. Biological samples for this study were in part supplied by our collaborators who obtained the original ethical permission for the sampling in African countries. Supplementary Table 3 contains the ethics reference numbers and local and Swedish permission details associated with these samples. Saliva samples were obtained using an Oragene DNA OG-500 kit. DNA was extracted using the prepIT L2P extraction protocol.

Primers were designed to amplify the full mtDNA genome using the program Snapgene Viewer (version 5.0.8). Eight sets of primers were tested for their ability to amplify the full mtDNA genome. The best set is shown in Supplementary Table 4. Barcoded primers were used and the full primer sequences can be found in Supplementary Table 5. A total of 1024 unique combinations of barcoded forward and reverse primers can be created using these 32 forward and 32 reverse primers, allowing the pooling of 1024 samples at the same time for one sequencing run. Using a unique barcode combination for every sample, we performed a PCR to amplify the whole mtDNA genome (30x (98 °C, 10 sec; 67 °C, 15 min); 4 °C ∞), (300 ng DNA, 2.4 nM primers, 200 μM of each dNTP, 1x PCR buffer and 1.25 U Takara GXL Taq in a total volume of 25 μl). Specificity of PCR products was confirmed on a 1% agarose gel and purified with AMPure XP beads (Beckman Colter). Concentrations of the cleaned PCR products were measured (Qubit, Broad Range kit) and samples were pooled (100 ng/ sample). Pools were purified with 0.5x volumes AMPure XP beads and eluted in 10 mM Tris-HCl, pH 8.5. Concentration of the cleaned pools was measured on the Qubit. The complete mtDNA genomes were sequenced in two separate sequencing runs using long-read sequencing technology on the PacBio Sequel II instrument at the National Genomics Infrastructure, Sci-LifeLab in Uppsala. The first batch contained 292 samples, and the second batch contained 1024 samples. From the 1316 newly sequenced samples, there were 16 technical duplicates and 12 samples that did not yield sufficient mtDNA reads, leaving 1288 samples for analysis.

### Haplogroup assignment

Demultiplexing of the sequencing data was performed by Uppsala Genome Center (UGC) at NGI-SciLifeLab using the SMRT analysis pipeline. As the reads span the beginning of the Revised Cambridge Reference Sequence (rCRS), the full mtDNA sequence reads were mapped to a duplicated rCRS to create BAM files. Two bioinformatics methods to call variants were compared; DeepVariant[91] in combination with bcftools consensus (version 1.12) and GATK HaplotypeCaller[92]. All scripts are available at https://github.com/imkelankheet/Full_mitochondrial_genomes.git. The two methods were compared using vcftools (–gzdiff –diff-site). Differences among the called nucleotides between these two methods were investigated and DeepVariant was chosen for downstream analyses due to its superior reliability in detecting insertions and deletions. Average sequencing coverage was determined per sample using samtools depth (samtools version 1.12). Mitochondrial haplogroups were assigned using HaploGrep3[74]. To ensure the independence between HaploGrep3 quality scores and coverage, we conducted a comparison by plotting the HaploGrep3 quality score against the coverages for all the samples in the second sequencing run (Supplementary Fig. 30).

### Database of full mtDNA sequences

A comparative database with 3612 publicly available full mtDNA sequences was assembled (Supplementary Data 1). Only haplogroups associated with African populations were considered (L, M1 and U6). Sampling locations were retrieved from the publications or through correspondence with the author. In cases where precise sampling locations were not explicitly

provided, but the population was known, an approximate location was deduced. This estimation involved identifying the midpoint of the known inhabited area from public databases such as the linguistic collection of Glottolog, serving as a practical proxy for the sampling location. Sequences were acquired using batch download from NCBI. The Human Genome Diversity Project (HGDP), Simons Genome Diversity Project (SGDP) and 1000 genomes project (KGP) fasta sequences were acquired from CRAM files using *bcftools mpileup, bcftools call* and *vcfutils.pl*. Mitochondrial haplogroups were assigned ex novo using HaploGrep3[74] and samples were assigned a maternal ancestry based on their haplogroup, using previously published literature on haplogroup-ancestry associations (see Supplementary Table 1). All individuals were assigned to language group if language information was available. A distinction was made between Afro-Asiatic, Nilo-Saharan, Bantu, non-Bantu Niger-Congo and Mande, Khoisan and Afrikaans (an Indo-European language of original Dutch origin). A full list of all accession numbers used for the database can be found in Supplementary Data 1.

## Haplogroup frequency maps
Haplogroup frequencies were computed based on the complete mtDNA sequence database and plotted on a map of Africa using the Kriging method[93]. As sites with a low number of individuals will bias our frequency distribution maps, we combined sites that were within one degree latitude and longitude in distance. Thereby, the number of unique sites was reduced from 356 to 220. This was done to increase the sample size and thereby gain more accuracy about the haplogroup composition. Merged sites with fewer than 10 individuals were removed from the analysis; this resulted in 112 different sites. Mitochondrial haplogroup frequencies were calculated for these 112 different sites and their distribution was plotted on a map of Africa using the Kriging method using Globemapper from Blue Marble Geographics (version 22.0) and Surfer from Golden Software (version 12.8.1009).

## Nucleotide diversity maps
We computed nucleotide diversity ($\pi$) within the complete mtDNA sequence database, with a requirement of a minimum of 10 individuals for calculation. To ensure as many coordinates to reach this minimum number of individuals of a certain haplogroup per coordinate, we created binned sites within three degrees in latitude and longitude. This broader binning in comparison to the haplogroup frequency analysis was necessary because nucleotide diversity calculations are restricted to individuals carrying a specific haplogroup, whereas haplogroup frequencies are calculated on all haplogroups together. Therefore, a larger bin size for the nucleotide diversity helps to ensure that we do not lose too many sites due to insufficient sample sizes. Nucleotide diversity was then determined for haplogroups L0a, L0d, L1c, L2a, and L3e at each of these binned coordinates. Nucleotide diversity was visualized on a map of Africa using Globemapper from Blue Marble Geographics (version 22.0) and Surfer from Golden Software (version 12.8.1009) using the Kriging method.

## Female effective population sizes through time
For visual representation of $N_e$ over time, Bayesian Skyline Plots (BSPs) were generated for various language groups: Nilo-Saharan, Afrikaans, Afro-Asiatic, Khoisan (all haplogroups), Khoisan (only L0k and L0d), Niger-Congo (Bantu speaking individuals not included) and Mande, Niger-Congo and Mande (300 Bantu speakers included), Bantu, North-Western Bantu, Eastern Bantu, West-Western Bantu and South-Western Bantu, as well as for various groups of individuals carrying Bantu-haplogroups among the North-Western Bantu, Eastern Bantu, West-Western Bantu and South-Western Bantu. Ninety-five individuals could not be assigned a language group due to the lack of information, and they were not used for the estimation of female $N_e$. Input files were prepared with BEAUTi using fasta files and subsequently processed with BEAST (version 1.8.4)[94] using the Markov Chain Monte Carlo (MCMC) sampling algorithm. Details regarding the number of MCMC iterations, the burn-in iterations discarded, and the

number of replicates for each language group are provided in Supplementary Table 6. A general time-reversible (GTR) substitution model was used, as suggested by ModelTest-NG version 1.0.0. A strict clock model was applied with mutation rate ($\mu$) of $2.285 \times 10^{-8}$ per site per year as used in Maier et al.[60], which is based on an average of mtDNA mutation rates surveyed in the literature[75,87,95–99]. These estimates are based on the human-chimp divergence time, aDNA data and phylogeographic differences. A Coalescent Bayesian Skyline model was chosen, with four groups and UPGMA starting tree. The number of iterations to be discarded as burn-in was determined by examining when the likelihood had stabilized. Resulting log and trees files were merged using the program LogCombiner (version 2.6.7) and all ESS values were checked to be above 200. Visualization of the BSP was done in Tracer (version 1.7.2)[100] and R (version 1.2.5033).

## Phylogenetic trees
Bayesian phylogenetic trees (Maximum Clade Credibility trees) were constructed using mtDNA sequences associated with African haplogroups. The Neanderthal mtDNA genome (GenBank accession number: AM948965) was included as an outgroup. One representative sample with the highest HaploGrep3 score was selected for each African haplogroup at the two classification level e.g. "L0d". The tree construction was performed using BEAST (version 1.8.4)[94]. The BEAST settings mirrored those described for the BSPs, employing a birth-death model. Five replicates of 100 million iterations were executed, with a burn-in of 10 million iterations. The resulting log and tree files were merged using LogCombiner (version 2.6.7). Subsequently, all effective sample size (ESS) values were verified to be above 200. Tree annotation was accomplished using TreeAnnotator from the BEAST package, and the final tree was visualized using the ggtree package[101] in R. For each haplogroup leaf, data on region and linguistic information were computed and visualized using R. Moreover, we generated Maximum Clade Credibility trees using BEAST for L0a, L0d, L0f, L0k, L1c, L2a, and L3e. Trees were annotated using TreeAnnotator (version 2.6.7) and these consensus trees were visualized in FigTree (version 1.4.4). L0d was added as an outgroup for all haplogroups, except for L0d itself, for which L0k was chosen as an outgroup. The TMRCA was taken from the visualized trees. The TMRCA was recalibrated using the correction proposed by Soares et al. to correct for purifying selection[75].

## Phylogenetic networks
Median Joining Networks were constructed using Network v.10.2 (www.fluxus-engineering.com). Maximum parsimony post-processing was performed using the Steiner maximum parsimony algorithm. The epsilon parameter was left unchanged and transversions were weighted three times the weight of transitions. Networks were visualized in Network Publisher and individuals were colored based on their regional affiliation.

## Declaration of generative AI and AI-assisted technologies in the writing process
During the preparation of this work the authors used ChatGPT 4o in order to assist with language and grammar checks. After using this tool, the authors reviewed and edited the content as needed and take full responsibility for the content of the publication.

## Statistics and reproducibility
All statistical analyses and computational procedures were performed using established software and standard parameters as described in the corresponding sections. These softwares include DeepVariant, bcftools, samtools, HaploGrep3, BEAST, and Tracer. Details of each analysis, including software versions and parameter settings, are provided in the Methods and Supplementary Tables. For sequencing-based analyses, technical duplicates were included to assess reproducibility, and 16 duplicates were processed independently from PCR reaction through sequencing and variant calling. Average sequencing coverage per sample was calculated using samtools depth to ensure consistent data quality across samples. Haplogroup assignments were independently verified by comparing results obtained

from two variant calling pipelines (DeepVariant and GATK Haplotype-Caller). Analyses of haplogroup frequencies and nucleotide diversity were based on a minimum of 10 individuals per geographic bin to ensure reliable estimation of population-level parameters. For Bayesian Skyline Plot analyses and phylogenetic trees, convergence and sufficient sampling were confirmed by verifying that all effective sample size (ESS) values exceeded 200 in Tracer.

## Reporting summary

Further information on research design is available in the Nature Portfolio Reporting Summary linked to this article.

## Data availability

All data generated or analyzed during this study is included in this published article, its supplementary information files and publicly available repositories. Source data for the figures can be found in Supplementary Data 2–10. The generated mitochondrial data are available as FASTA for academic research use through the NCBI database with accession numbers PV558957–PV560130, and PX394655 and PX394656. FASTQ files are available for academic research use through ENA with study number PRJEB108938, accession numbers ERS29408145–ERS29409320. All other data are available from the corresponding author on reasonable request.

## Code availability

Scripts are freely available on Zenodo (https://doi.org/10.5281/zenodo.18847351) and GitHub (https://github.com/Schlebusch-lab/Full_mitochondrial_genomes).

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

## Acknowledgements

We are grateful to all the individuals who voluntarily participated in this research. The authors would like to acknowledge support of the National Genomics Infrastructure (NGI)/Uppsala Genome Center and UPPMAX for providing assistance in massive parallel sequencing and computational infrastructure. Work performed at NGI/Uppsala Genome Center has been funded by RFI/VR and Science for Life Laboratory, Sweden. The computation and data handling were enabled by resources provided by the National Academic Infrastructure for Supercomputing in Sweden (NAISS) at UPPMAX partially funded by the Swedish Research Council through grant agreement no. 2022-06725. This project was supported by funding to CS from the European Research Council (ERC) under the European Union's Horizon 2020 research and innovation program (grant agreement No. 759933), the Knut and Alice Wallenberg foundation, the Leakey foundation and the Erik Philip Sorensson foundation. VČ was funded by Czech Academy of Sciences award Praemium Academiae. CB was supported by the URPP "Evolution in Action" of the University of Zurich and by the NCCR Evolving Language, Swiss National Science Foundation Agreement #51NF40_180888. Genetic sampling in the DRC as part of the BantuFirst project was supported by the European Research Council (ERC) (grant No. 724275) to KB. ML was supported by the Swedish Research Council grant 2020-04789. MdN was supported by European Union's Horizon 2020 research and innovation programme under the Marie Skłodowska-Curie grant agreement No 791695 (TimeAdapt).

## Author contributions

I.L. contributed to study design, performed experiments, carried out data analysis, and prepared the manuscript. A.C. and C.J. performed experimental work. C.T.R. performed mapping of the FASTQ files. A.E.R.S. and M.dN. contributed to discussions on study design. S.P., L.M., G.K., J.P.D., V.C., M.d.C., P.E., E.Pr., E.Po., V.Č., S.T.G., P.H., L.B., F.L.M.F., Z.G.T., W.A.M., M.d.J., H.S., K.B., and C.M.S. contributed by providing samples and conducting fieldwork to obtain them. K.B. and C.B. contributed through discussions and provided input and review of the manuscript. M.L. and H.M. contributed through discussions and provided feedback on the manuscript. C.M.S. conceived and designed the study, obtained the funding, supervised the project, and contributed to discussions and interpretation of results. All authors read and approved the final manuscript.

## Funding

## Competing interests

The authors declare no competing interests.

## Additional information

[1]Human Evolution, Department of Organismal Biology, Uppsala University, Uppsala, Sweden. [2]Department of Immunology, Genetics and Pathology, Uppsala Genome Center, Uppsala University, Uppsala, Sweden. [3]National Genomics Infrastructure, SciLifeLab, Uppsala, Sweden. [4]Department of Medical Biochemistry and Micro-biology, Uppsala University, National Bioinformatics Infrastructure Sweden, Uppsala, Sweden. [5]CBGP, INRAE, CIRAD, IRD, Institute Agro, University of Montpellier, Montpellier, France. [6]UGent Centre for Bantu Studies (BantUGent), Department of Languages and Cultures, Ghent University, Ghent, Belgium. [7]Institut Langage, Université de Mons, Mons, Belgium. [8]Research Institute for Languages and Cultures of Asia and Africa, Tokyo University of Foreign Studies, Tokyo, Japan. [9]Institut Supérieur Pédagogique de la Gombe, Kinshasa, Democratic Republic of the Congo. [10]Department of Biochemistry, Genetics and Microbiology, University of Pretoria, Pretoria, South Africa. [11]Biotechnology Platform, Agricultural Research Council, Onderstepoort, Pretoria, South Africa. [12]Department of Health Science and Technology, University of Aalborg, Aalborg, Denmark. [13]Archaeogenetics Laboratory, Institute of Archaeology of the Czech Academy of Sciences, Prague, Czech Republic. [14]School of Geography, Archaeology and Environmental Studies, University of the Witwatersrand, Johannesburg, South Africa. [15]Department of History and Archaeology, University of Lomé, Lomé, Togo. [16]Department of History and African Civilizations, Faculty of Arts, University of Buea, Buea, Cameroon. [17]Department of Microbial Sciences and Genetics, Addis Ababa University, Addis Ababa, Ethiopia. [18]Department of Anthropology and Archaeology, University of South Africa, Pretoria, South Africa. [19]Division of Human Genetics, School of Pathology, Faculty of Health Sciences, University of the Witwatersrand, Johannesburg, South Africa. [20]Academy of Science of South Africa, Pretoria, South Africa. [21]Department of Life and Environmental Sciences, University of Cagliari, Monserrato, CA, Italy. [22]Department of Evolutionary Biology and Environmental Studies, University of Zurich, Zurich, Switzerland. [23]Palaeo-Research Institute, University of Johannesburg, Johannesburg, South Africa. [24]Center for the Human Past, Department of Organismal Biology, Uppsala University, Uppsala, Sweden. [25]Science for Life Laboratory, Department of Organismal Biology, Uppsala University, Uppsala, Sweden. ✉e-mail: carina.schlebusch@ebc.uu.se

