## [Transparent Peer Review file · Communications Biology]

Revisiting the African mtDNA Landscape Through Complete Mitochondrial Genomes

Corresponding Author: Professor Carina Schlebusch

Version 0:

Reviewer comments:

Reviewer #1

(Remarks to the Author)

The manuscript by Schlebusch et al. is a timely concise and comprehensive overview of the maternal genetic diversity across the sub-Saharan African continent, based on 4,888 complete mtDNA genomes from 14 countries. A total of 1,288 of these mtDNA genomes are new, and were comprehensively analysed together with 3,600 publicly available African mitogenomes. Given the groundbreaking insights into human evolution contributed by the various mtDNA-based studies before the advent of complete human genomes, that were correctly referred to in the manuscript, this complete and updated revisit of the African mtDNA landscape deserves to be published, and will be very useful for the discipline and the broad audience interested in human evolution.

I have one major concern: the authors used a strict molecular clock with a mutation rate, not allowing for correction due to purifying selection, which affects significantly ages estimated for the human mtDNA phylogeny. For that reason, age estimates included in the manuscript are considerably young for the oldest TMRCA. Authors should discuss this issue in detail in the manuscript and try to estimate the TMRCA for all humans in their dataset by correcting for purifying selection. Otherwise, all the analyses were conducted correctly and in detail.

Reviewer #2

(Remarks to the Author)

The manuscript focuses on maternal ancestry in African populations, a subject of central relevance for understanding human evolution. It is commendable that the authors concentrate on mitochondrial DNA (mtDNA), which provides insights into maternal population history over extended timeframes due to its maternal mode of inheritance and lack of recombination. A key strength is the inclusion of a large and diverse dataset of complete mitochondrial genomes, with substantial additions from previously understudied groups. This addition is essential for reconstructing regional maternal demographic histories. The authors show a strong grasp of the continent's genetic, linguistic, and cultural diversity, integrating this into their interpretations. The study also introduces a more efficient mtDNA sequencing method, offering a valuable tool for future research.

The study has some drawbacks. Addressing them will improve the research.

General comments and suggestions.

A large body of research shows that mitochondrial DNA (mtDNA) mutations accumulate non-linearly rather than remain constant (Ho et al., 2005; Loogväli et al., 2009; Soares et al., 2009). The result is that there are increasingly more mutations per unit of time closer to the present day. Therefore, using any linear rate, including that from Maier et al., 2022, is problematic. Nevertheless, this rate is possibly close to a faster rate of recent periods (Soares et al., 2009). However, the difference becomes especially pronounced on deeper branches. African lineages are often deep, as seen in this and other studies. Using an equally fast rate for all periods likely explains the substantially younger time to the most recent common ancestor (TMRCA) of all anatomically modern human (AMH) mtDNA diversity estimated here and by Maier et al., 2022. I do not discuss the AMH–Neanderthal mtDNA TMRCA here and recommend the authors do the same unless an appropriate outgroup is included. For an accurate AMH–Neanderthal TMRCA, one must include a chimpanzee or Denisovan mtDNA sequence to correctly assign substitutions to the Neanderthal lineage versus the branch leading from the AMH–Neanderthal MRCA to the MRCA of all AMH. Otherwise, BEAST cannot distinguish which substitutions should be assigned to each

branch, and the resulting estimate represents a mathematical average rather than an actual divergence time. To my knowledge, BEAST does not offer a built-in solution for addressing mutation rate non-linearity. However, there is at least one workaround that can be applied post hoc: using the scaling correction formula proposed by Soares et al., 2009 on the age estimates obtained from BEAST analyses. Given the substantial computational demands, there is no need to rerun all analyses using the mutation rate specified by Soares et al., 2009. Instead, authors may consider the following simpler alternative. First, it is necessary to rescale the BEAST-derived ages to match the linear mutation rate of Soares et al., 2009. Then, the authors should apply the correction formula from the same study. Other approaches may exist to account for non-linearity in Bayesian analyses, but I am unaware of them.

The current analysis relies on three-symbol haplogroup nomenclature, which inherently corresponds to deep branches and, thus, to older events. In contrast, the linguistic processes under consideration are comparatively recent. For some groups, for example, Khoe-San, using three symbol levels is appropriate because sub-branches of L0d or L0k are also mainly connected to these groups. L0d in Indo-European-speaking Afrikaans is a much later development and broadly signals Khoe-San history. For other groups, using more recent branches for frequency comparisons is a better alternative. For example, haplogroup L2a is prevalent across many African populations; however, its initial spread (corresponding to the L2a node) predates major language expansions. More derived L2a subclades may better capture signals of migration events that coincide with linguistic expansions. Hence, it would become possible to distinguish whether sub-branches of L2a in non-Bantu speakers reflect admixture from Bantu migrations or predate them in situ. The authors appear to be aware of these nuances; a comparable point is made in the final paragraph of the Discussion, where the Bayesian Skyline results for non-Bantu Niger-Congo and Mande speakers are interpreted.

Another suggestion is to indicate the source of uncertainty for node ages. Do they reflect only the phylogenetic uncertainty or incorporate published confidence intervals for the mutation rate?

Specific comments and suggestions.

Lines 94-95. According to the sentence, M, N, and R are sequences. That is incorrect because these are macro-haplogroups or haplogroups.

Lines 111-116. The sentence is long, which makes it hard to follow. Please improve this part.

Line 215. How many mitogenomes have eventually been removed after filtering with a 30x threshold? Do the numbers in this paragraph represent sample numbers before or after filtering? Please make that clear in the text.

Lines 359-364. Median-joining networks are problematic in several respects. First, they allow present-day sequences to occupy internal nodes, which contradicts the assumption that all internal nodes represent unsampled ancestors. Second, unlike Bayesian phylogenetic approaches, this method does not account for demographic changes (e.g., population expansions or contractions) that can influence the shape of coalescent trees. Consequently, the topology produced by a median-joining network often reflects methodological limitations rather than genuine population history. The median-joining analysis adds little since Bayesian phylogenetic reconstructions for L3e1 and L3e2 (using the same dataset) are already available and provide reliable estimates of relationships and divergence times. It is better to remove the analysis and its associated discussion (lines 607–617).

Lines 474-477. Please be more cautious here. While the difference is somewhat clear for Eastern Bantu, it is unclear for other Bantu speakers. With such large credible intervals, asserting a clear temporal distinction is premature.

Lines 479-482, 620-624. Likewise, there is insufficient support to conclude that haplogroup L2a expanded later than all haplogroups together.

Lines 483-486. In Supplementary Figure 26, the N_e trajectory for L1c does not exhibit a pronounced expansion or contraction. Possible slight changes in N_e are insufficient to characterize as a bottleneck or expansion.

Lines 626-628. This observation appears to apply to L3e only.

Figure-related comments and suggestions

Please indicate the exact number of sequences analyzed in each Bayesian Skyline plot. This can be added directly to the panels or noted in each figure legend.

The panels in Figure 4 use different Y-axis limits, which may mislead readers when comparing population sizes across groups. Please unify the lower and upper limits and maintain a consistent scale across all panels. This adjustment will facilitate a direct visual comparison of N_e trajectories. The issue is common in Figure 5 and supplementary figures with many Bayesian Skyline panels in a plot.

Figure 5. Panel D already depicts parts of the information in panels B and C. Additional information in Panel B is the median curve for all haplogroups of non-Bantu Niger-Congo speakers. Panel B also visualizes a comparison of all haplogroups' median curves. The intent of panel C is to compare different regions by haplogroup L3e median curves. These are the same curves depicted in panel D differently. Please move the sub-regional curves from panel B into the Supplementary Materials to improve readability. In Figure 5, panel B would show only Skyline curves for all non-Bantu Niger-Congo and Bantu speakers, each with corresponding credible intervals. Similarly, relocate panel C (haplogroup L3e) to the Supplementary Materials, ensuring credible intervals are included. Please include credible intervals if sub-regional comparisons are still desired in Figure 5

Supplementary Figure 29. Based on the number of L3b mitogenomes, Bayesian Skyline analysis is impossible not only for North-Western but also for West-Western Bantu. It is hard to imagine how the algorithm can estimate node groupings with such a low number of nodes. Anything below 30 is probably not worth conducting.

References

Simon Y. W. Ho, Matthew J. Phillips, Alan Cooper, Alexei J. Drummond, Time Dependency of Molecular Rate Estimates and Systematic Overestimation of Recent Divergence Times, *Molecular Biology and Evolution*, Volume 22, Issue 7, July 2005, Pages 1561–1568, <https://doi.org/10.1093/molbev/msi145>

Loogväli EL, Kivisild T, Margus T, Villems R (2009) Explaining the Imperfection of the Molecular Clock of Hominid Mitochondria. PLOS ONE 4(12): e8260. <https://doi.org/10.1371/journal.pone.0008260>

Soares P, Ermini L, Thomson N, Mormina M, Rito T, Röhl A, Salas A, Oppenheimer S, Macaulay V, Richards MB. Correcting for purifying selection: an improved human mitochondrial molecular clock. Am J Hum Genet. 2009 Jun;84(6):740-59. <https://doi: 10.1016/j.ajhg.2009.05.001>

Reviewer #3

(Remarks to the Author)

I think this is an important addition of human mitochondrial sequences from under-sampled areas of Africa. I like the encyclopedic description of the distribution and age of variation (and of population size and diversity) across Africa. The supplemental files were very useful. I found the choice of analyses to be acceptable. The data quality seems quite good, with avg of >900x coverage. Presumably the samples could also be used for nuclear data as well in the future. This is a very straightforward study that provides important data for understanding African variation and diasporas.

Version 1:

Reviewer comments:

Reviewer #1

(Remarks to the Author)

The authors addressed all my comments. I agree with the publication as it is.

Reviewer #2

(Remarks to the Author)

I thank the authors for their responses to my comments and for the revisions made to the manuscript. Compared with the initial submission, the study has improved, and many of my comments have been addressed. However, I would like to request a justification in the manuscript for the continued use of the linear mtDNA mutation rate from Maier et al. (2022) as the primary rate in the manuscript, given that the authors acknowledge the non-linearity of the mtDNA mutation process. This rate underlies the reported branch age estimates, the visualization of the phylogenetic trees, and the skyline plot reconstructions. An explicit explanation of why this linear rate is retained as the main framework would help readers better assess the assumptions of these analyses.

I also note that, in most cases, the uncorrected slow rate from Soares et al. (2009) appears unnecessary and redundant, and its added value is unclear.

Finally, regarding the network analyses, the authors describe them as a “visual aid” in their rebuttal. I disagree with this characterization. In practice, the networks function as a visual aid only when one accepts the methodological limitations previously outlined in my review. I do not request their removal from the manuscript; however, if they are retained, their limitations and restricted interpretative value should be stated explicitly.

Version 2:

Reviewer comments:

Reviewer #2

(Remarks to the Author)

I thank the authors for their response. I have no further comments.

Reviewers' comments:

Reviewer #1:

The manuscript by Schlebusch et al. is a timely concise and comprehensive overview of the maternal genetic diversity across the sub-Saharan African continent, based on 4,888 complete mtDNA genomes from 14 countries. A total of 1,288 of these mtDNA genomes are new, and were comprehensively analysed together with 3,600 publicly available African mitogenomes. Given the groundbreaking insights into human evolution contributed by the various mtDNA-based studies before the advent of complete human genomes, that were correctly referred to in the manuscript, this complete and updated revisit of the African mtDNA landscape deserves to be published, and will be very useful for the discipline and the broad audience interested in human evolution.

We thank the reviewer for their positive and encouraging comments on our manuscript. We are pleased that they find the work to be a valuable contribution to the field.

I have one major concern: the authors used a strict molecular clock with a mutation rate, not allowing for correction due to purifying selection, which affects significantly ages estimated for the human mtDNA phylogeny. For that reason, age estimates included in the manuscript are considerably young for the oldest TMRCA. Authors should discuss this issue in detail in the manuscript and try to estimate the TMRCA for all humans in their dataset by correcting for purifying selection.

We thank the reviewer for raising this important point. We agree that applying a strict molecular clock without correcting for purifying selection can lead to underestimated ages, particularly for the deepest nodes of the mtDNA phylogeny. To address this, we have now incorporated additional TMRCA estimates that account for purifying selection. Specifically, we present (i) estimates based on the original mutation rate of 2.285×10^{-8} , (ii) recalculated estimates using a slower mutation rate of 1.665×10^{-8} , and (iii) recalibrated estimates using the correction proposed by Soares *et al.* (2009). As expected, the corrected values are consistently older, with the largest differences observed for the deepest nodes, including the TMRCA of all modern humans (see line 252–258, line 925-928, and Supplementary Table 6).

Otherwise, all the analyses were conducted correctly and in detail.

Reviewer #2:

The manuscript focuses on maternal ancestry in African populations, a subject of central relevance for understanding human evolution. It is commendable that the authors concentrate on mitochondrial DNA (mtDNA), which provides insights into maternal population history over extended timeframes due to its maternal mode of inheritance and lack of recombination. A key strength is the inclusion of a large and diverse dataset of complete mitochondrial genomes, with substantial additions from previously understudied groups. This addition is essential for reconstructing regional maternal demographic histories. The authors show a strong grasp of the continent's genetic, linguistic, and cultural diversity, integrating this into their interpretations. The study also introduces a more efficient mtDNA sequencing method, offering a valuable tool for future research.

We thank the reviewer for their thoughtful and encouraging comments on our manuscript. We are pleased that they appreciate both the dataset and methodological aspects of the study, as well as the integration of genetic, linguistic, and cultural perspectives.

The study has some drawbacks. Addressing them will improve the research.

General comments and suggestions.

A large body of research shows that mitochondrial DNA (mtDNA) mutations accumulate non-linearly rather than remain constant (Ho et al., 2005; Loogväli et al., 2009; Soares et al., 2009). The result is that there are increasingly more mutations per unit of time closer to the present day. Therefore, using any linear rate, including that from Maier et al., 2022, is problematic. Nevertheless, this rate is possibly close to a faster rate of recent periods (Soares et al., 2009). However, the difference becomes especially pronounced on deeper branches. African lineages are often deep, as seen in this and other studies. Using an equally fast rate for all periods likely explains the substantially younger time to the most recent common ancestor (TMRCA) of all anatomically modern human (AMH) mtDNA diversity estimated here and by Maier et al., 2022.

We agree with the reviewer that using a constant mutation rate likely underestimates the ages of deeper branches, particularly in African lineages. We have addressed this concern in detail in our response to two points below, where we recalculated and recalibrated TMRCA following the approach of Soares et al. (2009). In brief, these corrections result in consistently older age estimates (see line 252–258, line 925-928, and Supplementary Table 6).

I do not discuss the AMH–Neanderthal mtDNA TMRCA here and recommend the authors do the same unless an appropriate outgroup is included. For an accurate AMH–Neanderthal TMRCA, one must include a chimpanzee or Denisovan mtDNA sequence to correctly assign substitutions to the Neanderthal lineage versus the branch leading from the AMH–Neanderthal MRCA to the MRCA of all AMH. Otherwise, BEAST cannot distinguish which substitutions should be assigned to each branch, and the resulting estimate represents a mathematical average rather than an actual divergence time.

We thank the reviewer for pointing this out. Following the suggestion, we have removed the AMH–Neanderthal split time from the Results section and from Supplementary Table 6. We now only report divergence times that can be robustly estimated with the dataset and outgroup used.

To my knowledge, BEAST does not offer a built-in solution for addressing mutation rate non-linearity. However, there is at least one workaround that can be applied post hoc: using the scaling correction formula proposed by Soares et al., 2009 on the age estimates obtained from BEAST analyses. Given the substantial computational demands, there is no need to rerun all analyses using the mutation rate specified by Soares et al., 2009. Instead, authors may consider the following simpler alternative. First, it is necessary to rescale the BEAST-derived ages to match the linear mutation rate of Soares et al., 2009. Then, the authors should apply the correction formula from the same study. Other approaches may exist to account for non-linearity in Bayesian analyses, but I am unaware of them.

We thank the reviewer for this valuable suggestion. In response, we now report three different TMRCA estimates to explicitly account for mutation rate uncertainties and the effects of purifying selection. Specifically, we present (i) estimates based on the original mutation rate of 2.285×10^{-8} , (ii) recalculated estimates using a slower mutation rate of 1.665×10^{-8} , and (iii) recalibrated estimates applying the correction formula proposed by Soares *et al.*, 2009 to account for purifying selection. As expected, the corrected estimates are consistently older, with the largest differences observed for the deepest nodes, including the TMRCA of all modern humans. These additions have been incorporated into the Methods section (line 925-928), into the main text (line 252–258), and into Supplementary Table 6.

The current analysis relies on three-symbol haplogroup nomenclature, which inherently corresponds to deep branches and, thus, to older events. In contrast, the linguistic processes under consideration are comparatively recent. For some groups, for example, Khoe-San, using three symbol levels is appropriate because sub-branches of L0d or L0k are also mainly connected to these groups. L0d in Indo-European-speaking Afrikaans is a much later development and broadly signals Khoe-San history. For other groups, using more recent branches for frequency comparisons is a better alternative. For example, haplogroup L2a is prevalent across many African populations; however, its initial spread (corresponding to the L2a node) predates major language expansions. More derived L2a subclades may better capture signals of migration events that coincide with linguistic expansions. Hence, it would become possible to distinguish whether sub-branches of L2a in non-Bantu speakers reflect admixture from Bantu migrations or predate them in situ. The authors appear to be aware of these nuances; a comparable point is made in the final paragraph of the Discussion, where the Bayesian Skyline results for non-Bantu Niger-Congo and Mande speakers are interpreted.

Another suggestion is to indicate the source of uncertainty for node ages. Do they reflect only the phylogenetic uncertainty or incorporate published confidence intervals for the mutation rate?

We thank the reviewer for this comment. We agree that the use of three-symbol haplogroup nomenclature often corresponds to deep branches that predate the more

recent linguistic processes in our study. In the revised manuscript, we have clarified this issue (line 608-613) in the section on haplogroup L3e, where we now explicitly note that because L3e is a much older haplogroup (36.1 kya, CI: 30.0–42.4 kya), it is unlikely that all L3e lineages were involved in the Bantu expansion. Instead, only specific subhaplogroups of L3e are likely to have been associated with the earliest Bantu-speaking groups, and further work will be needed to identify which subclades capture this signal.

Regarding the source of uncertainty for node ages, we have clarified this in Supplementary Table 6, where we now state: “Reported 95% confidence intervals reflect the phylogenetic uncertainties.”

Specific comments and suggestions.

Lines 94-95. According to the sentence, M, N, and R are sequences. That is incorrect because these are macro-haplogroups or haplogroups.

We thank the reviewer for pointing this out. We adapted the text accordingly: “Haplogroup L3 gave rise to all mtDNA sequences outside the African continent (belonging to haplogroups M, N and R).” (line 95-96).

Lines 111-116. The sentence is long, which makes it hard to follow. Please improve this part.

We adapted this sentence by breaking it up into smaller parts and changing the order of the sentence (line 112-118).

Line 215. How many mitogenomes have eventually been removed after filtering with a 30x threshold? Do the numbers in this paragraph represent sample numbers before or after filtering? Please make that clear in the text.

We thank the reviewer for pointing this out as this revealed the fact that some low-coverage samples had been included in previous analyses. We removed those samples and re-ran all affected analyses and made sure only samples with coverage over 30X were included. These re-analyzed results did not change any of the main conclusions of the manuscript. The total number of samples sequenced was 1316 (292 in the first run and 1024 in the second run). We had 130 samples with coverage lower than 30x which were filtered out, and 10 technical duplicates. Removing low-coverage samples and the technical duplicates leaves 1176 samples for analysis. We have clarified this in the text (line 218-220).

Lines 359-364. Median-joining networks are problematic in several respects. First, they allow present-day sequences to occupy internal nodes, which contradicts the assumption that all internal nodes represent unsampled ancestors. Second, unlike Bayesian phylogenetic approaches, this method does not account for demographic changes (e.g., population expansions or contractions) that can influence the shape of coalescent trees. Consequently, the topology produced by a median-joining network often reflects methodological limitations rather than genuine population history. The median-joining analysis adds little since Bayesian phylogenetic reconstructions for L3e1 and L3e2 (using the same dataset) are already available and provide reliable estimates of relationships and

divergence times. It is better to remove the analysis and its associated discussion (lines 607–617).

We appreciate the reviewer's comment regarding median-joining networks. We are aware of the methodological limitations of this approach - specifically, that present-day sequences can occupy internal nodes and that demographic changes are not explicitly accounted for. Consequently, we did not rely on these networks for our main inferences, which are based on Bayesian phylogenetic reconstructions.

However, we used median-joining networks primarily as a visual aid to illustrate certain aspects of the Bantu expansion. For example, terminal nodes often reflect geographic patterns, with Southern African individuals appearing at the network edges for some subhaplogroups and Central African individuals for others, consistent with signals of back-migrations, possibly during recent migration processes such as the the Mfecane. We believe this visualization helps contextualize our findings without over-interpreting the network topology.

That said, if the reviewer feels strongly that these analyses are unnecessary, we are willing to remove the median-joining networks and the associated discussion.

Lines 474-477. Please be more cautious here. While the difference is somewhat clear for Eastern Bantu, it is unclear for other Bantu speakers. With such large credible intervals, asserting a clear temporal distinction is premature.

We appreciate this comment and agree that the wide credible intervals limit our ability to make strong temporal distinctions. We have therefore rephrased the passage to be more cautious (line 481-486). Instead of asserting clear sequential differences, we now state that the results seem to indicate that the expansion of North-Western, West-Western, and South-Western Bantu groups occurred later than that inferred with L3e individuals, while Eastern Bantu groups may represent an exception. However, given the large and overlapping credible intervals, these observations should be interpreted with caution.

Lines 479-482, 620-624. Likewise, there is insufficient support to conclude that haplogroup L2a expanded later than all haplogroups together.

We have revised the text to make the interpretation of the timing of the L2a expansion more cautious. We also added a sentence noting that broad and overlapping credible intervals in the BSPs mean that these apparent temporal differences should be interpreted with caution. This clarifies that we cannot conclusively state that L2a expanded later than other haplogroups, but only highlighting possible interpretations. (line 489-494 and line 634-640).

Lines 483-486. In Supplementary Figure 26, the Ne trajectory for L1c does not exhibit a pronounced expansion or contraction. Possible slight changes in Ne are insufficient to characterize as a bottleneck or expansion.

We agree with the reviewer and changed the text to say there are slight indications of expansions and contractions, noting at the same time that the confidence intervals are high (line 494-497).

Lines 626-628. This observation appears to apply to L3e only.

We agree that the effect is more pronounced in L3e, but we observe the phenomenon in both L3e and L2a. We have clarified this in the text accordingly (line 644-645).

Figure-related comments and suggestions

Please indicate the exact number of sequences analyzed in each Bayesian Skyline plot. This can be added directly to the panels or noted in each figure legend.

We have now added the number of sequences to Figure 4, Figure 5, Supplementary Figures 26-30. Supplementary figure 12 and 14 already reported the number of sequences in their legends. Moreover, we added the number of sequences to the legend of Supplementary Figure 24 and 25.

The panels in Figure 4 use different Y-axis limits, which may mislead readers when comparing population sizes across groups. Please unify the lower and upper limits and maintain a consistent scale across all panels. This adjustment will facilitate a direct visual comparison of N_e trajectories. The issue is common in Figure 5 and supplementary figures with many Bayesian Skyline panels in a plot.

We thank the reviewer for this helpful suggestion, as it makes the figures clearer. We have unified the Y-axis limits across all panels in Figure 4, Figure 5D, and the supplementary figures containing multiple Bayesian Skyline plots (Supplementary Figures 26-30).

Figure 5. Panel D already depicts parts of the information in panels B and C. Additional information in Panel B is the median curve for all haplogroups of non-Bantu Niger-Congo speakers. Panel B also visualizes a comparison of all haplogroups' median curves. The intent of panel C is to compare different regions by haplogroup L3e median curves. These are the same curves depicted in panel D differently. Please move the sub-regional curves from panel B into the Supplementary Materials to improve readability. In Figure 5, panel B would show only Skyline curves for all non-Bantu Niger-Congo and Bantu speakers, each with corresponding credible intervals. Similarly, relocate panel C (haplogroup L3e) to the Supplementary Materials, ensuring credible intervals are included. Please include credible intervals if sub-regional comparisons are still desired in Figure 5

We appreciate this comment. We agree that there is some overlap between panels B, C, and D, but we prefer to keep the figure as it is. Each panel is intended to highlight a different comparison: panel B serves to compare the effective population size trajectories of different language groups analyzed; panel C shows the convergence of the BSPs for haplogroup L3e

across regions; and panel D illustrates that the onset of expansion in L3e subgroups is earlier compared to the expansion pattern seen when considering all haplogroups together. Although related, these perspectives address distinct points we wish to emphasize, and we feel the current structure conveys these comparisons most effectively. In addition, we have created a more detailed version of panel B, including confidence intervals, and included it in the Supplementary Materials as Supplementary Figure 26.

Supplementary Figure 29. Based on the number of L3b mitogenomes, Bayesian Skyline analysis is impossible not only for North-Western but also for West-Western Bantu. It is hard to imagine how the algorithm can estimate node groupings with such a low number of nodes. Anything below 30 is probably not worth conducting.

We agree with the reviewer that the plot of West-Western Bantu looked not trustworthy and removed it from this Figure, which is Supplementary Figure 30 in the updated manuscript.

References

Simon Y. W. Ho, Matthew J. Phillips, Alan Cooper, Alexei J. Drummond, Time Dependency of Molecular Rate Estimates and Systematic Overestimation of Recent Divergence Times, *Molecular Biology and Evolution*, Volume 22, Issue 7, July 2005, Pages 1561–1568, <https://doi.org/10.1093/molbev/msi145>

Loogväli EL, Kivisild T, Margus T, Villems R (2009) Explaining the Imperfection of the Molecular Clock of Hominid Mitochondria. *PLOS ONE* 4(12): e8260. <https://doi.org/10.1371/journal.pone.0008260>

Soares P, Ermini L, Thomson N, Mormina M, Rito T, Röhl A, Salas A, Oppenheimer S, Macaulay V, Richards MB. Correcting for purifying selection: an improved human mitochondrial molecular clock. *Am J Hum Genet.* 2009 Jun;84(6):740-59. <https://doi:10.1016/j.ajhg.2009.05.001>

Reviewer #3:

I think this is an important addition of human mitochondrial sequences from under-sampled areas of Africa. I like the encyclopedic description of the distribution and age of variation (and of population size and diversity) across Africa. The supplemental files were very useful. I found the choice of analyses to be acceptable. The data quality seems quite good, with avg of >900x coverage. Presumably the samples could also be used for nuclear data as well in the future. This is a very straightforward study that provides important data for understanding African variation and diasporas.

We are grateful for the reviewer's positive assessment that our study provides valuable data for understanding African genetic variation and diasporas.

Reviewers' comments:

Reviewer #1:

The authors addressed all my comments. I agree with the publication as it is.

We thank the reviewer for the positive evaluation of our revised manuscript and for their careful assessment throughout the review process.

Reviewer #2:

I thank the authors for their responses to my comments and for the revisions made to the manuscript. Compared with the initial submission, the study has improved, and many of my comments have been addressed. However, I would like to request a justification in the manuscript for the continued use of the linear mtDNA mutation rate from Maier et al. (2022) as the primary rate in the manuscript, given that the authors acknowledge the non-linearity of the mtDNA mutation process. This rate underlies the reported branch age estimates, the visualization of the phylogenetic trees, and the skyline plot reconstructions. An explicit explanation of why this linear rate is retained as the main framework would help readers better assess the assumptions of these analyses.

We now include an explicit justification for retaining the linear mitochondrial mutation rate from Maier et al. (2022) as the primary calibration framework. Specifically, we explain that this rate is consistent with the assumptions of BEAST and allows direct comparison with other recent studies that apply the same calibration. We also clarify that, although the non-linearity of the mitochondrial mutation process is well known, we use the Soares et al. (2009) correction to illustrate its effects. As now noted in the manuscript, the recalibrated ages are consistently older, and the uncorrected estimates should therefore be interpreted as minimum ages. The new text has been added to the manuscript, highlighted in red (lines 253–266).

I also note that, in most cases, the uncorrected slow rate from Soares et al. (2009) appears unnecessary and redundant, and its added value is unclear.

We have now removed the uncorrected slow rate from Soares et al. (2009) from the manuscript (see Supplementary Table 6).

Finally, regarding the network analyses, the authors describe them as a “visual aid” in their rebuttal. I disagree with this characterization. In practice, the networks function as a visual aid only when one accepts the methodological limitations previously outlined in my review. I do not request their removal from the manuscript; however, if they are retained, their limitations and restricted interpretative value should be stated explicitly.

We have now expanded the description of the network analyses in the Discussion to explicitly acknowledge their methodological limitations and restricted interpretative value. Specifically, we added the following sentence (line 639-641):

“Networks do have certain limitations that should be kept in mind, such as the possibility that present-day sequences may occupy internal nodes and that demographic processes are not explicitly accounted for.”

This statement clarifies the restricted interpretative scope of the network analyses, in line with the reviewer’s suggestion.